# Cancer co-opts differentiation of B-cell precursors into macrophage-like cells

Chen Chen[1], Bongsoo Park[2], Emeline Ragonnaud[1], Monica Bodogai[1], Xin Wang[1], Le Zong[2], Jung-Min Lee[3], Isabel Beerman [2,4] & Arya Biragyn [1,4] ✉

We have recently reported that some cancers induce accumulation of bone marrow (BM) B-cell precursors in the spleen to convert them into metastasis-promoting, immunosuppressive B cells. Here, using various murine tumor models and samples from humans with breast and ovarian cancers, we provide evidence that cancers also co-opt differentiation of these B-cell precursors to generate macrophage-like cells (termed B-MF). We link the transdifferentiation to a small subset of CSF1R[+] Pax5[Low] cells within BM pre-B and immature B cells responding to cancer-secreted M-CSF with downregulation of the transcription factor Pax5 via CSF1R signaling. Although the primary source of tumor-associated macrophages is monocytes, B-MFs are phenotypically and functionally distinguishable. Compared to monocyte-derived macrophages, B-MFs more efficiently phagocytize apoptotic cells, suppress proliferation of T cells and induce FoxP3[+] regulatory T cells. In mouse tumor models, B-MFs promote shrinkage of the tumor-infiltrating IFNγ[+] CD4 T cell pool and increase cancer progression and metastasis, suggesting that this cancer-induced transdifferentiation pathway is functionally relevant and hence could serve as an immunotherapeutic target.

The role of B cells in cancer remains poorly understood, as their presence is positively and negatively associated with the disease outcome. Even in the same murine tumor models, different types of B cells promote or retard cancer escape, thereby affecting the progression of B16-F10 melanoma in C57BL/6 mice[1,2] and lung metastasis of orthotopic 4T1 breast cancer cells in BALB/c mice[3,4]. At least some cancer-promoting functions of B cells can be attributed to their regulatory subsets, such as TGFβ[+] CD25[+] Bregs (tBregs), which support lung metastasis by inducing FoxP3[+] Tregs or educating MDSCs via targeting TGFβRII[5,6]. The generation and activation of B cells and Bregs, in turn, is regulated by cancer-secreted factors, such as B lymphocyte stimulator (BLyS/BAFF), thymic stromal lymphopoietin (TSLP), colony-stimulating factors (M-CSF, GM-CSF, and G-CSF), and lipid mediators such as 5-lipoxygenase (5-LO) metabolites[5,7–11]. For example, we recently reported that cancer remotely downregulates CXCR4 and α4β1 integrin on pro-B and pre-B cells in the bone marrow

(BM) using TSLP to cause their premature emigration and accumulation in the spleen[11]. It is to convert these cells into tBregs by targeting PPARα signaling with 5-LO metabolites[8]. We also found that TSLP from cancers prepares the metastasis "soil", such as inducing expression of CCL17 in the lungs to recruit CCR4[+] cancer cells and their protector CCR4[+]FoxP3[+] Tregs and Th2-skewed CD4[+] T cells[7,12]. Cancer-secreted or induced M-CSF and GM-CSF promotes differentiation and survival of cancer-promoting myeloid suppressive cells and tumor-associated macrophages (TAM) from BM monocytes[9,10]. However, their role in the differentiation of lymphocytes remains poorly understood. Although bifurcation of myeloid and lymphoid lineage from multipotent progenitors occurs before specialization of B-cell progenitors in BM and monocytes give rise to macrophages, B-cell precursors appear to retain the macrophage-differentiation potential, as they can transdifferentiate into macrophages after forced expression or deletion of single transcription factors[13,14]. In

[1]Immunoregulation Section, Laboratory of Immunology and Molecular Biology, National Institute on Aging, Baltimore, MD, USA. [2]Epigenetics and Stem Cell Aging Unit, Translational Gerontology Branch, National Institute on Aging, Baltimore, MD, USA. [3]Women's Malignancies Branch, Center for Cancer Research, National Cancer Institute, Bethesda, MD, USA. [4]These authors jointly supervised this work: Isabel Beerman, Arya Biragyn. ✉e-mail: biragyna@mail.nih.gov

naïve mice, a small subset of biphenotypic pro-B cells (CD19[+]B220[+]CD16/32[++]CD11b[+]) with non-rearranged B-cell receptor (BCR) genes is recently reported to acquire macrophage phenotypes, albeit at very low levels[15]. The biological consequence of this rare event and whether cancers affect the B-cell-to-macrophage trans-differentiation remain unknown.

Here, we report that cancers transdifferentiate the bona fide BM B-cell precursors, including Csf1R[+]Pax5[Low] pre-B and immature IgM[+] B cells, into TAM (termed B-MF) using M-CSF. Unlike monocyte-derived TAM[10], cancers use B-MF to mediate escape and metastasis via suppressing antitumor IFNγ[+]CD4[+] T cells. This does not appear to be a mouse-specific phenomenon, as B-MF-like cells and their transcriptional signature can be detected in patients with breast and ovarian cancers and in published scRNA sequence data of human cancers.

## Results

### TAM expresses B-cell markers

We previously reported that some cancers mobilize BM B-cell precursors in the spleen[11] to convert them into TGFβ[+] tBregs[5,6,8]. Microarray transcription profiling of these B cells in the spleen of BALB/c mice with orthotopic 4T1.2 breast cancer (a model for human triple-negative breast cancer[16]) surprisingly revealed significant upregulation of macrophage-associated genes, such as *CD68*, *Csf1r* (encodes CSF1R), *Cebpb* (CCAAT Enhancer binding protein beta), *Cebpg* (CCAAT Enhancer binding protein gamma), *Ccl2* (CCL2), and *Csf1* (M-CSF) (Fig. 1a). Given that CEBPB and CSF1R play essential roles in defining macrophage fate[13,17] and that the biphenotypic B-cell progenitors and B1 B cells can generate macrophages in mice[15,18], we tested whether cancer induces the macrophage-like cells from pre-B cells by FACS evaluating tumor-infiltrating CD19[+] B cells (TIB) and macrophages (TAM, based on F4/80[+]CD11b[+])[19] in B-cell sufficient (WT) and deficient BALB/c mice (μMT, where B cells do not differentiate beyond pro-B cells[20]) with 4T1.2 cancer. WT mice contained small numbers of F4/80[+]CD11b[+] TIB and CD19[+] and CD79a[+] TAM, which were almost undetectable in μMT mice (Fig. 1b, gating strategy is in Supplementary Fig. 1a). Compared with CD79[−] TAM (presumably bona fide macrophages), the CD79[+] TAM expressed CD20, IgM, and IgD and significantly upregulated F4/80, CD11b, CD206, IL4Rα, and binding to Filipin (a fluorescent polyene antibiotic that detects cellular free cholesterol[21]) (Supplementary Fig. 1a, b). These cells (hereafter referred to as B-MF) were also found in primary tumors of C57BL/6 mice with s.c. MC38 colon cancer and in the tumor microenvironment (peritoneum) of mice with spontaneous ovarian Mogp cancer, but again were almost lost in μMT and J$_H$T mice (Fig. 1c and Supplementary Fig. 1c–f), where B cells cannot differentiate beyond pro-B cells[20,22]. Immunohistochemistry staining for CD19 and CD68 (a marker of macrophages and mononuclear phagocytes[23]) also revealed a small number of CD19[+] cells within CD68[+] myeloid cells and clusters of CD19[+] B cells in the primary tumors of WT mice with 4T1.2 and MC38 cancers (Fig. 1d and Supplementary Fig. 1g, h). We also evaluated B-MF in Mb1-Cre/Rosa-EYFP crossed (Mb1-EYFP) mice with or without peritoneal ID8 ovarian cancer, where Mb1-dependent Cre-recombinase causes B-cell-exclusive expression of EYFP (enhanced yellow fluorescent protein)[24]. Compared with tumor-free Mb1-EYFP (naïve) mice, the peritoneum of ID8 cancer-bearing mice was significantly enriched in B2 B cells and B-MF expressing EYFP (Fig. 1e, f and Supplementary Fig. 2a, b). The B-MF also upregulated the expression of CD274 and TGFβ/LAP (Supplementary Fig. 2c), two immunoregulatory factors[5,25,26]. In contrast, regardless of the tumor-bearing or naïve states of mice, these cells were only present at a small frequency in the spleen and LN (Supplementary Fig. 2a, b). In sum, we concluded that B-MF are derived from B cells, which cancer either expands, or de novo differentiates in the tumor microenvironment.

## Cancer induces B-cell transdifferentiation

Because B-MF could be misinterpreted as trogocytosis or cell fusion[27,28], we performed a series of B-cell differentiation experiments using highly FACS-purified CD19[+]B220[+] B cells (Lin[−], >99% purity, Supplementary Fig. 3a) from BM of naïve mice. The cells were cultured in a conditioned medium (CM) of 4T1.2 cancer cells (4T1.2-CM) to FACS-evaluate surface expression of B-cell and macrophage markers. B cells gradually became CD11b[+]F4/80[+] while downregulating CD19 and some CD79a expression by 7–8 days of incubation in 4T1-CM (Fig. 2a, b and Supplementary Fig. 3b). After 14-day culture, the cells remained IgM[+]CD11b[High]F4/80[High] but further decreased CD19 and CD79a (Supplementary Fig. 3c). From here on, to capture these cells in "transition", we used 7-day incubation for experimental timepoints, unless specified otherwise. To further examine these cells, we performed single-cell Imagestream FACS analysis and confirmed the Mb1-EYFP[+]/CD79[+] cells expressed CD20, F4/80, and CD11b and were larger in size than bona fide B cells (Fig. 2c and Supplementary Fig. 3d). B-MF also acquired additional macrophage features, such as the ability to adhere to plastic and phagocytize fluorochrome-labelled *E. coli* (Supplementary Fig. 3e). By culturing FACS-purified B-cell subsets in 4T1.2-CM, we linked the B-MF generation to BM B-cell precursors and immature IgM[+] B cells (collectively termed as BMBP), but not to peripheral B cells in naïve mice, including splenic transitional, follicular (FOB), or marginal zone (MZB) B cells (Fig. 2e, f; Supplementary Fig. 3f; gating strategy in Supplementary Fig. 3g). Similarly, CM from almost every type of cancer cells, except B16-F10 melanoma, induced the generation of B-MF from naïve mouse BMBP and the immortalized 70z/3 pre-B-cell line after 7 and 30 days of culture, respectively (Fig. 2g and Supplementary Fig. 3h–k). B cells cultured in the control cRPMI medium did not generate B-MF (Fig. 2d, g). To rule out trogocytosis/cell fusion, we performed in vitro and in vivo B-MF conversion assays using CD45.1 or CD45.2 alloantigen-expressing C57BL/6 mice. First, we cultured a mixture of FACS-purified EYFP[+] BMBP from CD45.2[+] mice and BM monocytes from CD45.1[+] mice in 4T1.2-CM for 7 days to generate B-MF and monocyte-derived macrophages (Mo-MF). While a small fraction (1–3%) of cells co-expressed CD45.2 and CD45.1 (presumably a result of trogocytosis or cell fusion), the majority of B-MF and Mo-MF only expressed their parental single alloantigen, CD45.2 or CD45.1 (Fig. 2h and gating strategy and purity, Supplementary Fig. 4a, b), implying they were not derived from trogocytosis/cell fusion. To confirm this conclusion in vivo, we transferred FACS-purified EYFP[+] BM B cells from naïve CD45.2[+] mice into the peritoneum of CD45.1[+] mice with a 21-day-old ID8 tumor (Fig. 2i). After 7 days, FACS analysis of the transferred cells revealed that only a very small fraction of them co-expressed CD45.1 and CD45.2/EYFP (presumably due to trogocytosis/cell fusion), while the majority of EYFP[+] B-MF did not express CD45.1 (Fig. 2i, j and Supplementary Fig. 4c). Taken together, we concluded that cancer generates TAM by transdifferentiating BMBP in addition to their hitherto known source, monocytes[10].

### B-MF transcription profiles are distinct from Mo-MF

To understand the nature of these B-MF, we compared their phenotypes to that of Mo-MF (generated in 4T1.2-CM, as described above). While both B-MF and Mo-MF highly upregulated F4/80 and CD11b but not DC and granulocyte markers (Supplementary Fig. 5a), adhered to plastic, showed similar cell size as peritoneal macrophages (Supplementary Fig. 5b), only B-MF expressed the B-cell-specific markers (CD79a and IgM, Fig. 3a, b and Supplementary Fig. 5c). In mRNA microarray analysis, B-MF and Mo-MF shared expression of numerous macrophage-related genes regardless of the origin (Supplementary Fig. 5d–f and Supplementary Data file 1), although principal component analysis (PCA) clearly separated the two cell types (Fig. 3c and Supplementary Data file 2). B-MF expressed higher levels of genes involved in fatty acid metabolism, oxidative phosphorylation, cell cycle, steroid-cholesterol biosynthesis, and downregulated expression

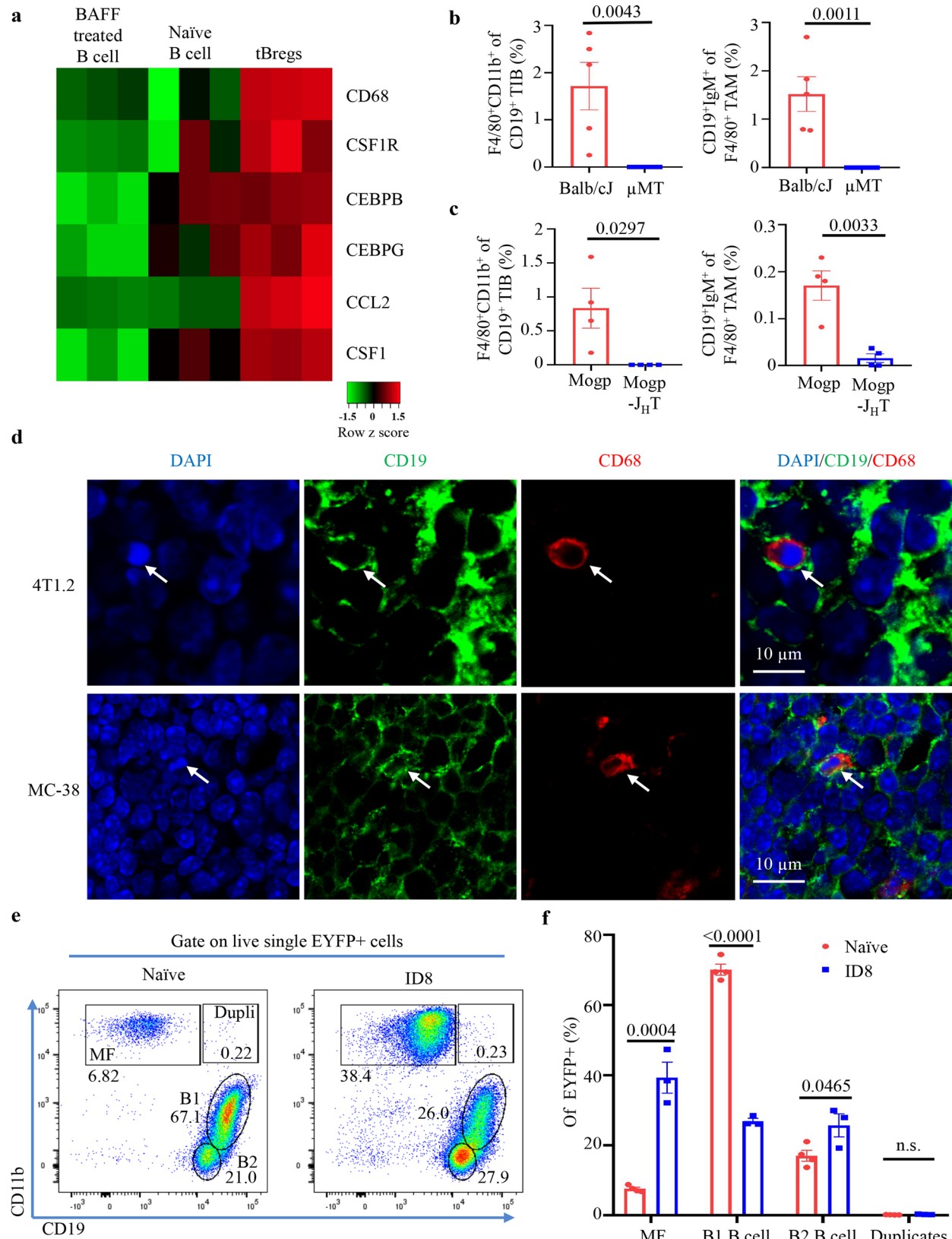

of pro-inflammatory and IFNγ response genes (Fig. 3d, e). While Mo-MF were enriched for a M1-like transcription profile, M2-skewing was more pronounced in B-MF (Fig. 3e and Supplementary Fig. 5g). The unique transcription profiles were also confirmed in single-cell RNA sequencing (scRNA-seq) of B-MF (10,563) and Mo-MF (10,235) cells, with UMAP clustering identifying mostly separate cell clusters of the two

cell types (Fig. 3f and Supplementary Data file 3). We distinguished 12 cell clusters with the Leiden algorithm, using shared nearest neighbor (SNN) in PCA space and identified the key genes establishing the six clusters accounting for the majority of single cells (Fig. 3g, h and Supplementary Fig. 5i). B-MF appeared to be more phagocytic than Mo-MF, as they markedly upregulated *Mrc1* (encodes CD206, Fig. 3i).

**Fig. 1 | B-cell and macrophage marker co-expressing cells present in the TIB and TAM from cancer microenvironment. a** mRNA microarray heatmap showing macrophage-specific gene expression in tBregs as compared with naïve or BAFF-treated B cells from spleen (*n* = 3 mice). Scale bar is for expression z-score. **b, c** FACS staining frequency (Mean ± SEM) of expression of macrophage (MF, F4/80⁺CD11b⁺) and B-cell (CD19⁺IgM⁺) surface markers in, respectively, TIB and TAM from the primary tumors of BALB/CJ and µMT mice with orthotopic 4T1.2 breast cancer (*n* = 5 for BALB/CJ, *n* = 6 for µMT **b**), and in the peritoneum of C57BL/6 and J_HT mice with Mogp cancer (Mogp and Mogp-J_HT, respectively, *n* = 4, **c**). *P*-values in **b** (*P* = 0.0043 and *P* = 0.0011 are for indicated cells in BALB/CJ vs µMT) and **c** (*P* = 0.0297, *P* = 0.0033 are for indicated cells in Mogp vs Mogp-J_HT) were calculated using two-tailed unpaired *t*-test. Gating strategy is shown in Supplementary Fig. 1a. **d** Representative immunohistochemistry staining of primary tumors from BALB/c and C57BL/6 mice with 4T1.2 breast cancer (upper panel) and MC38 colon cancer (lower panel) detects CD19⁺CD68⁺ cells withing B cells (CD19⁺) and myeloid phagocytes (CD68⁺, *n* = 3 mice per group). Scale bar is for 10 µm. **e, f** ID8 cancer-bearing MB1-EYFP mice significantly increase MB1-EYFP⁺ cells in the peritoneum with ID8 cancer as compared to that of naïve EYFP mice. Representative FACS plot and Mean frequency ± SEM (*n* = 3 mice) of MF, B1 B cells (B1) and B2 B cells withing EYFP cells are shown in **e** and **f**, respectively. *P*-values in **f** were calculated using two-tailed unpaired *t*-test (MF *P* = 0.0004; B1 B cell *P* < 0.0001; B2 B cell *P* = 0.0465 Naïve vs ID8). Consistent with marked loss of CD19 in B-MF, only negligible frequency of CD19⁺F4/80⁺CD11b⁺EYFP⁺ cells (Dupli or duplicates) are detected in both naïve and ID8 cancer-bearing mice. Results in **b**, **c**, and **e** were independently reproduced at least three times. From here on, Error bars are for SEM.

As in microarray analysis (Fig. 3d), expression of genes for oxidative phosphorylation were more upregulated in B-MF than Mo-MF (clusters 0, 4 and 6, Supplementary Fig. 5h), consistent with their M2-skewing[29]. Given the unique transcriptional signatures of the B-MF, we next examined scRNA profiles of TAM purified from four different mice with 4T1.2 cancer (Fig. 3j). To identify potential in vivo B-MF, we used signature genes identified from in vitro-generated macrophages (Fig. 3h) and noted three clusters (0, 6, and 8) with robust expression of genes identified in B-MF (Fig. 3j, Supplementary Fig. 5i–k). Cluster 8 also strongly overlapped with a mixed macrophage population (Cluster 4, Fig. 3g and Supplementary Fig. 5i), suggesting that only clusters 0 and 6 most likely represent B-MF. We also examined the expression of three genes robustly expressed in B-MF, and they were also expressed in TAM clusters 0 and 6 (Fig. 3k). In contrast, expression of key Mo-MF genes was mostly found in clusters 1, 3, 4, and 5 (Supplementary Fig. 5j, k), suggesting that the two macrophages retain traceable and different transcription profiles in vivo.

## Cancer generates B-MF to suppress antitumor CD4⁺ T cells

To confirm the differences between the two macrophages at the functional levels, we quantified their proliferation, phagocytosis of red fluorescent protein (RFP)-tagged apoptotic ID8 cells, and intracellular cholesterol. Only B-MF readily incorporated BrdU (pulsed on day 6 and tested on day 7 of the culture) and expressed higher levels of Ki67 (Fig. 4a and Supplementary Fig. 6a, b). Although the two macrophages phagocytized fluorochrome-labeled apoptotic cancer cells (Fig. 4b, c) and contained elevated levels of cellular cholesterol (Fig. 4d, e), both these features were significantly upregulated in B-MF compared to Mo-MF per cell-to-cell comparisons (Fig. 4c, e). Similarly, CD79⁺ TAM exhibited markedly higher Filipin binding than CD79⁻ TAM (Supplementary Fig. 1a–f). Because B-MF and CD79⁺ TAM also significantly upregulated TGFβ/LAP and PD-L1 (Supplementary Fig. 1a–f and Supplementary Fig. 2c) and lipid accumulation in TAM associated with suppression of anticancer CD8⁺ T cells[30], we wondered whether these cells promote tumor progression via regulating the activity of T cells. To test this possibility, first, we performed in vitro T-cell suppression assay[5] by culturing B-MF or Mo-MF with eFluor450-labeled naïve mouse T cells stimulated with anti-CD3/CD28 Abs and IL-2 at various effector: target ratios for 4 days. The B-MF, but not Mo-MF, significantly inhibited the proliferation of naïve mouse CD4⁺ T cells and, a lesser extent, CD8⁺ T cells in a dose-dependent manner (Fig. 4f and Supplementary Fig. 6c, d). Second, we performed a 5-day Treg conversion assay[5] by culturing B-MF and Mo-MF with naïve mouse FACS-purified CD25⁻CD4⁺ T cells in the presence of anti-CD3/CD28 Abs and IL-2. B-MF more efficiently induced the generation of FoxP3⁺Tregs than Mo-MF (Supplementary Fig. 6e). Next, we tested whether B-MF reverses a retarded tumor progression in µMT mice, which we previously linked to a lack of B cells[6]. µMT C57BL/6 and BALB/c mice with subcutaneous (s.c.) B16-F10 melanoma (*n* = 10–12/group) or 4T1.2 breast cancer (*n* = 12–14/group), respectively, were intravenously (i.v.) transferred with in vitro-generated B-MF (Fig. 4g). Compared with mock, B-MF significantly increased tumor weight in mice with melanoma (*p* < 0.01, Fig. 4h) and numbers of metastatic foci in the lungs of mice with 4T1.2 cancer (*p* < 0.05, Fig. 4i). FACS evaluation of their tumors surprisingly did not detect a difference in the presence of CD8⁺ and CD4⁺ T cells and FoxP3⁺ Tregs (Supplementary Fig. 6f, g). Instead, the B-MF transfer significantly decreased the frequency and numbers of IFNγ-expressing CD4⁺ T cells in both cancer models (Fig. 4j–m). In 4T1.2 tumors, B-MF also markedly decreased granzyme (Gr) B⁺ CD4⁺ cells (Supplementary Fig. 6g), which were implicated in tumor cell killing[31]. A separate transfer experiment with equal numbers of FACS-purified naïve mouse follicular B cells (FOB) or B-MF in µMT mice with 4T1.2 cancer (3 × 10⁵ cells/mouse, *n* = 5–7 mice per group) revealed that both cells comparably support lung metastasis (Supplementary Fig. 6h). Compared to B-MF, FOB upregulated numbers of CD4⁺ T cells but decreased frequency of IL10⁺ CD4⁺ T cells and GrB⁺ and Lamp1⁺ (cytolytic) CD8⁺ T cells in the tumor (Supplementary Fig. 6h), implying that the two cells support cancer independently and without reversal of B-MF to B cells. To confirm this, we performed a 3-day tracking experiment by i.v. transferring fluorochrome-labeled B-MF (500,000 cells/mouse) in µMT mice with 14-day 4T1.2 tumor. The majority of transferred cells were in the spleen and tumor (and less in dLN, Supplementary Fig. 6i). Per gram tumor, numbers of transferred B-MF were slightly less (about 7-fold) than that of 4T1.2 tumor in BALB/c mice and MC38 tumor in Mb1-YEFP mice (Supplementary Fig. 6k). Consistent with in vitro stability of B-MF phenotype (Fig. 2a, b and Supplementary Fig. 3b, c), the transferred cells were exclusively CD11b⁺F4/80⁺ (>98%, Supplementary Fig. 6j). Taken together, we concluded that cancers generate B-MF mostly to downregulate anticancer IFNγ⁺CD4⁺ T cells.

## Cancer mobilizes BMBP in the spleen to convert them to B-MF

Because cancer can mobilize BM pre-B cells in the spleen and tumor to generate tBregs[11], we tested whether this pool of BMBP is the source of B-MF. Compared with naïve mice, the total number of CD93⁺ BMBP was markedly decreased in BM but increased in the spleen as well as present in the tumors of mice with 4T1.2 and Mogp cancers (Fig. 5a, Supplementary Fig. 7a, b and not depicted). To link them to the generation of B-MF, we FACS-purified CD93⁺ and CD93⁻ BMBP from the spleen of mice with 4T1.2 cancer and naïve mice and cultured these cells in 4T1.2-CM. Only splenic CD93⁺, but not CD93⁻, BMBP from tumor-bearing mice generated B-MF, while B cells from spleens of naïve mice failed to do so regardless of CD93 expression (Supplementary Fig. 7c and Supplementary Fig. 3f), implying that cancer accumulates CD93⁺ BMBP in the spleen and tumor[11] to generate B-MF.

## Cancer targets CSF1R+CD93+ BMBP by secreting M-CSF

To understand the mechanism of the B-MF generation, we analyzed CM of cancer cells for secreted factors that could affect the differentiation of macrophages. M-CSF, a regulator of macrophage differentiation and survival[9,17], was among the factors that were highly increased in the cancer cells that induce B-MF (Fig. 5b). Conversely,

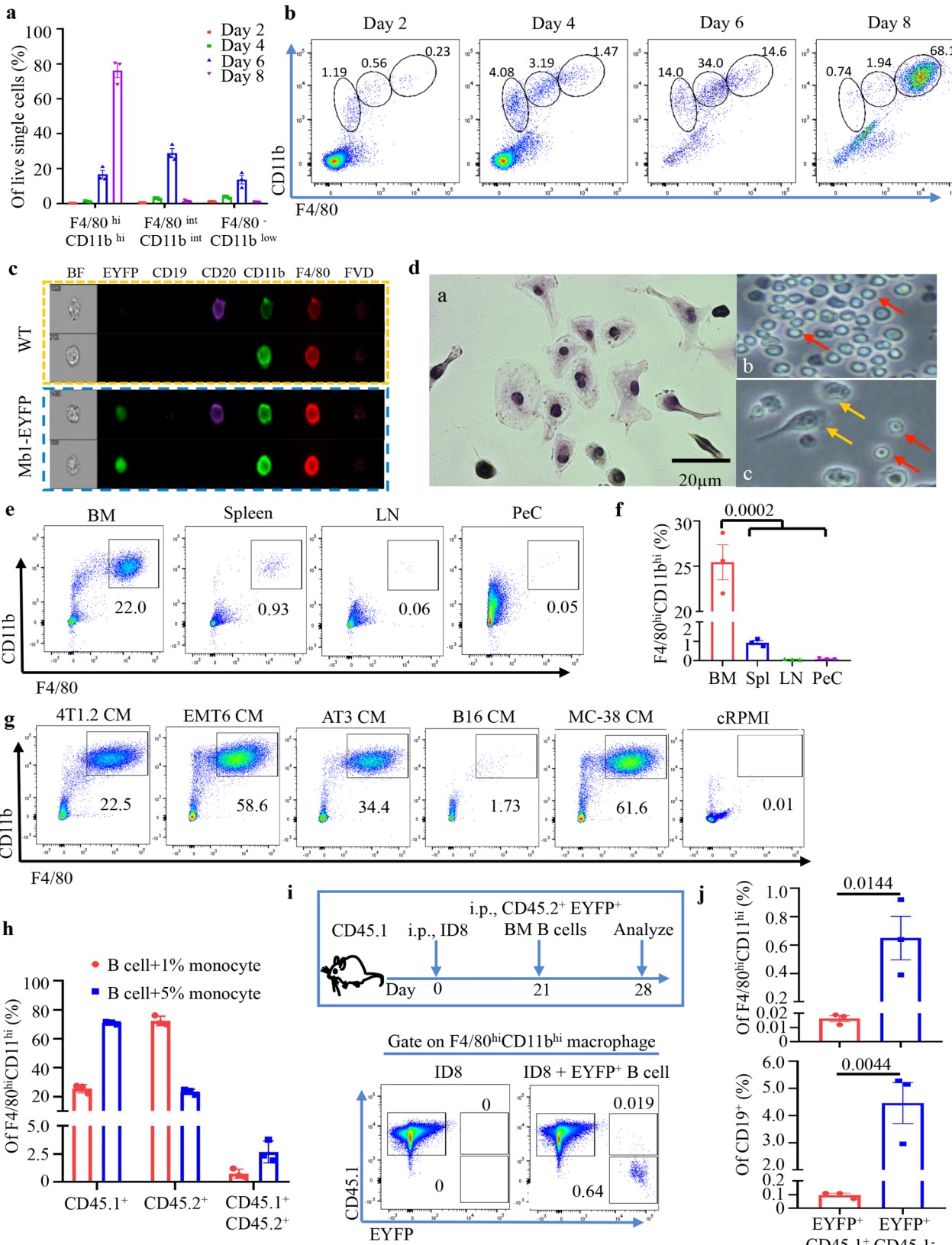

M-CSF was almost absent in CM from B16-F10 cells (Fig. 5b and Supplementary Fig. 7d, e), which did not induce the generation of B-MF (Fig. 2g and Supplementary Fig. 3h, i). Compared to naïve mice, serum M-CSF was also significantly upregulated in mice with 4T1.2 cancer (Supplementary Fig. 7f). Importantly, CD93⁺CD19⁺ BMBP of naïve mice expressed its cognate receptor CSF1R (about 15% of immature B cells,

6% of pre-B cells, and 1.5% pro-B cells, Supplementary Fig. 7g). In mice with 4T1 cancer, CSF1R⁺CD93⁺CD19⁺ BMBP were markedly reduced in BM but increased in the spleen (Fig. 5a and Supplementary Fig. 7g), consistent with their cancer-induced emigration from BM, as discussed above. To link these CSF1R⁺CD93⁺CD19⁺ BMBP to the generation of B-MF, highly FACS-purified CSF1R⁺ and CSF1R⁻ B cells from BM and

**Fig. 2 | Cancer induces differentiation of macrophages from BM B cells.**
**a**, **b** 4T1.2-CM cause transdifferentiation of BM B cells. Mean frequency ± SEM of FACS staining (**a** n = 3 mice; and **b** representative FACS plots) to show gradual upregulation of macrophage markers in indicated gates (circles, b) after culture in 4T1.2-CM. Representative Imagestream (**c**) and Giemsa staining (**d**) images of in vitro-generated B-MF from naïve WT (**c**, **d**) and Mb-EYFP⁺ mice (bottom panel, **c**) showing co-expression of B-cell and MF markers (**c**), larger size and adherence to plastic (**d**). Red and Yellow arrows are for nonadherent B cells and adherent B-MF, respectively (bright light images, **d**). Scale bar is of 20 μm. **e**, **f** B-MF are generated from BM B cells. Representative FACS plot (**e**) and quantification (Mean frequency ± SEM, **f**, n = 3 mice, P = 0.0002 BM vs spleen, LN and PeC) of F4/80ʰⁱCD11bʰⁱ B-MF converted from BM, spleen, inguinal LN, and PeC B cells of naïve BALB/c mice as in **e**. **g** Unlike control CM (B16-F10 cells or cRPMI), CM of indicated cancer cells induce the generation of B-MF from BM B cells. Numbers are for % of gated cells (B-MF, see Supplementary Fig. 3h for quantification results). **h**–**j** The B-MF generation is not a result of trogocytosis. **h** Frequency of CD45 isoforms from in vitro differentiation of BM CD45.2⁺ (EYFP⁺) B cells mixed with CD45.1⁺ monocytes (1% and 5%). Y-axis is for Mean frequency ± SEM of EYFP⁺/CD45.2⁺ and EYFP⁻/CD45.1⁺ cells within F4/80⁺CD11b⁺ cells, respectively (n = 3 mice, gating strategy in Supplementary Fig. 4a, b). **i** Schema of in vivo conversion of CD45.2⁺ (EYFP⁺) B cells in PeC of CD45.1⁺ mice with 21-day peritoneal ID8 tumor, and representative FACS plot (gaiting strategy is in Supplementary Fig. 4C). **j** Quantification of CD45.2 and CD45.1-expressing CD19⁺ (P = 0.0044) and F4/80⁺CD11b⁺ (P = 0.0144) cells (Mean frequency ± SEM, n = 3 mice). P-values in **f** and **j** were calculated using two-tailed unpaired t-test. Results were independently confirmed at least three times (**a/b**, **e/f**, and **g**) and twice (**c**, **d**, **h**, and **j**).

spleen of naïve mice were cultured in 4T1.2-CM. While CSF1R⁺ BM B cells readily generated B-MF, the BM CSF1R⁻ subset failed to do so (Fig. 5c). Consistent with the inability of splenic CD93⁺ B cells of naïve mice to generate B-MF (Supplementary Fig. 7c and Supplementary Fig. 3f), we failed to convert naïve mouse splenic B cells into B-MF regardless of CSF1R expression (Fig. 5c). We also cultured primary BM BMBP or 70z/3 cells with 4T1.2-CM in the presence or absence of neutralizing M-CSF antibody (Ab) or Ki20227, a specific inhibitor of c-Fms/CSF1R[32]. Both cells failed to generate B-MF upon M-CSF neutralization or CSF1R signaling inhibition (Fig .5d, e and Supplementary Fig. 7h, i). To rule out artifacts of in vitro assay, we created mice with conditional CSF1R deficiency in B cells (Mb1-CSF1Rᶠˡᵒˣ/ᶠˡᵒˣ mice, gating strategy in Supplementary Fig. 8). Unlike WT littermates or monocytes from Mb1-CSF1Rᶠˡᵒˣ/ᶠˡᵒˣ mice, the loss of CSF1R in BMBP significantly impaired the cancer CM-induced B-MF differentiation (Fig. 5f, g). Of note, the residual macrophage differentiation seen in Fig. 5f is presumably due to CSF1R expression preceding Mb1 expression, while Mb1-CSF1Rᶠˡᵒˣ/ᶠˡᵒˣ mice will have Csf1r deletion only in pro-B cells and onward.

Given that PAX5 is the key pro-B-cell factor that represses Csf1r and other myeloid lineage-specific genes[14,33], we reasoned that cancer decreases levels of this transcription factor using M-CSF. FACS staining confirmed that Pax5 was markedly decreased in BM CD93⁺ BMBP, particularly in CSF1R⁺ but not CSF1R⁻ subsets, from mice with 4T1.2 or Mogp cancers (Fig. 6a and Supplementary Fig. 9a–c). Importantly, Pax5 was also significantly decreased in BM CSF1R⁺ BMBP from naïve mice and 70Z/3 cells upon treatment with 4T1-CM or M-CSF (Fig. 6b, c and Supplementary Fig. 9b). As Pax5 deficiency alone is sufficient to render pro-B cells susceptible to myeloid differentiation[14], we concluded that cancer uses M-CSF to reduce expression of Pax5 in CSF1R⁺CD93⁺ BMBP and thereby promote macrophage differentiation.

To further understand the B-cell susceptibility towards macrophage conversion, we analyzed chromatin accessibility by performing ATAC-seq on CSF1R⁺ and CSF1R⁻ BMBP isolated from both BM and spleen of naïve mice. PCA clustering showed the most robust differences in chromatin profiles were driven by the location of the BMBP (BM vs spleen) regardless of CSF1R expression, driving the PC2 axis (blue and purple vs orange and green, Fig. 6d). The chromatin landscapes of the CSF1R⁺ and CSF1R⁻ BMBP isolated from BM (orange and green, Fig. 6d) also significantly differed from each other, driving the PC3 axis. We then examined the differentially accessible regions (DAR) between CSF1R⁺ and CSF1R⁻ cells isolated from the spleen or BM. Whereas comparisons between CSF1R⁺ and CSF1R⁻ cells from the spleen did not show any differences reaching our threshold for significance, confirming their close clustering on the PC3 axis; the BM CSF1R⁺ cells contained significantly more open chromatin than the BM CSF1R⁻ cells (Fig. 6d, e). These data suggest the BM CSF1R⁺ BMBP may have a more permissive chromatin environment, susceptible to macrophage-differentiation signals. As the splenic B cells and BM CSF1R⁻ cells were refractory to macrophage conversion (Fig. 5c), we looked at DARs with less accessibility in CSF1R⁻ compared to CSF1R⁺

BM cells (749 loci) in spleen cells to determine if these regions remain closed and potentially "lock in" the lymphoid lineage potential. Indeed, the overwhelming majority of regions with decreased accessibility in the BM CSF1R⁻ cells remained closed in the cells from the spleen (679 of 749). Evaluation of these consensus open regions found in BM CSF1R⁺ cells for potential transcription factor binding sites permitting macrophage differentiation showed significantly increased accessibility of ERG and RUNX1 sites (Fig. 6f). ERG is known to be expressed both in myeloid and lymphoid progenitor cells[34] and has particular importance in early hematopoietic progenitor cells as it binds to co-egulators such as RUNX and GATA[35]. RUNX1 regulates the growth and survival of macrophages via binding to promoter and enhancer regions of Csf1r and upregulating its expression[36]. Runx1 is also robustly expressed in early progenitor and myeloid-committed progenitor cells[37]. Thus, the increased accessibility to binding sites of both ERG and RUNX1 suggests a potentially more primitive, permissive chromatin state allowing for myeloid lineage transformation of the BM CSF1R⁺ cells.

## B-MF-generating CSF1R+ BMBP accumulate in humans with cancer

We recently reported that peripheral mobilization of BMBP also occurs in humans with breast cancer (BC)[11], suggesting the generation of B-MF. To test this possibility, we FACS evaluated peripheral blood (PB) of healthy donors (HD, n = 7) and patients with BC (n = 8). Compared with HD, PB of BC was markedly increased in CSF1R⁺ BMBP (Supplementary Fig. 9d), as we described in mice with cancer. Moreover, microarray transcription profiling of sort-purified B cells from PB of BC patients revealed that they significantly upregulated macrophage-associated genes, such as Cebpa, Marco, and Csf1r, as compared with B cells from HD (Fig. 6g). We also FACS evaluated B cells from PB of patients with ovarian cancer (OC, n = 5). Compared with HD, OC patients significantly increased CSF1R⁺ BMBP (Fig. 6h and Supplementary Fig. 9e) with upregulated expression of CD68 and LDLR (Fig. 6h), similar to mice with cancer. Using recently published scRNA-seq data of tumor-infiltrated immune cells from patients with breast cancer[38], we also found a macrophage cluster with overlapping signatures of B-MF-like cells (cluster 3, Fig. 6i) by examining genes with differential expression defined in murine in vitro-generated B-MF (cluster 0, Fig. 3h). In particular, cluster 3 was enriched for expression of EGR1, IER2, IER3, and SLC4OA1, which were major drivers of identity for murine in vitro-generated B-MF (Fig. 6i). Similarly, in the single-cell transcriptome data from human high-grade serous OC[39], we also detected the B-MF-like signature in macrophages (Cluster 0, Fig. 6j), although with a lesser overlap than in BC, further suggesting that human cancers can promote the B-cell transdifferentiation into macrophages.

## Discussion

BMBP undergo a series of subsequent and tightly regulated differentiation steps after their bifurcation from multipotent cells to

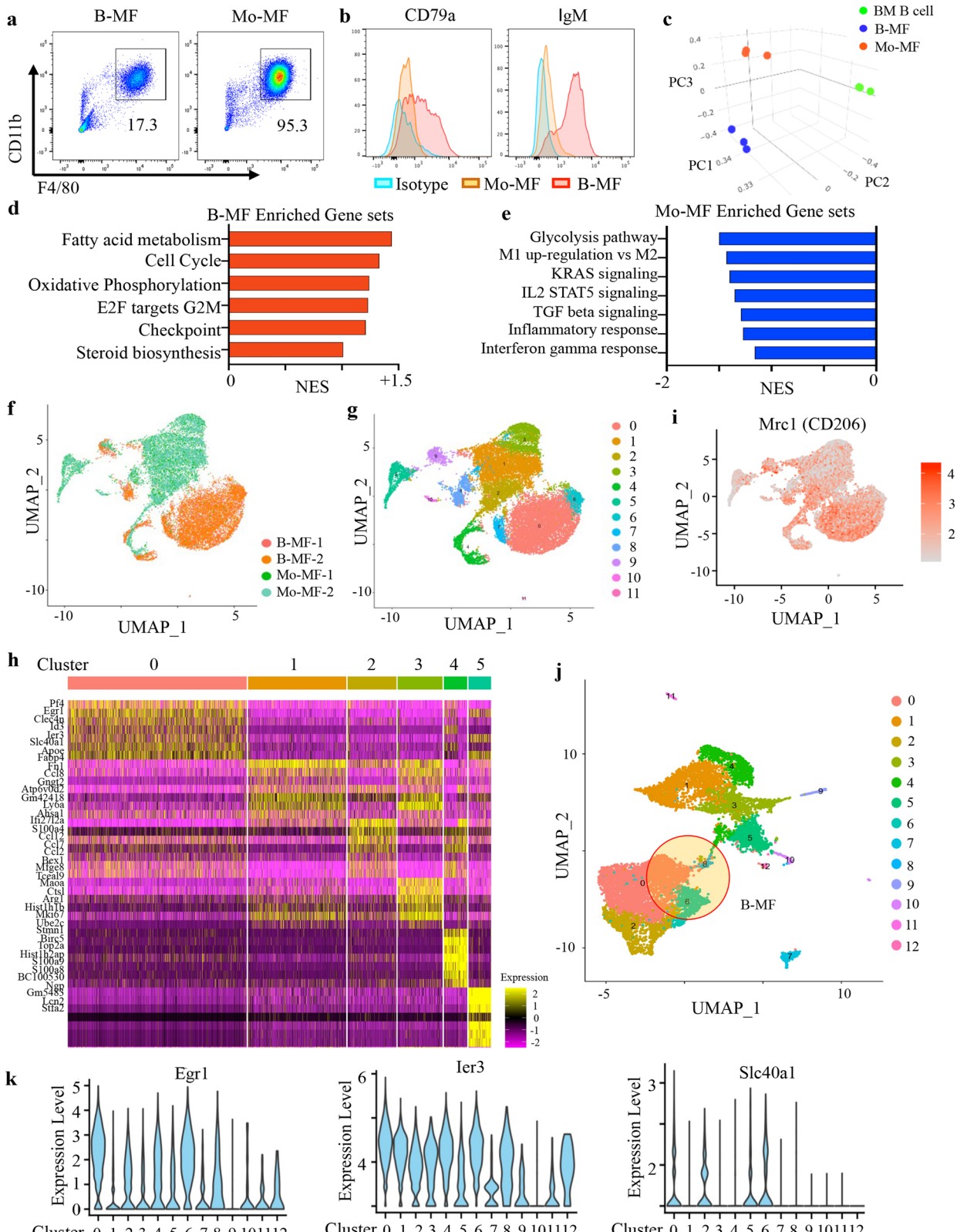

**Fig. 3 | Distinct gene expression profiles of B-MF and Mo-MF. a, b** Representative FACS plots (**a**) and histograms (**b**) of BM-MF and Mo-MF generated from BM B cells or monocytes after 7-day treatment with 4T1.2-CM. Numbers are for proportion of gated (F4/80^hiCD11b^hi) cells (**a**). **b** Shows expression of CD79a and IgM in B-MF (Red) and Mo-MF (Orange). **c** PCA plot of mRNA expression profiles generated from microarray data of sort-purified B-MF (Blue), Mo-MF (Orange) and BM B cells (Green) (*n* = 3 mice). **d, e** Bar plots of GSEA predicted pathways enriched in B-MF (**d**) or Mo-MF cells (**e**) from the Molecular Signature Database. **f, g** UMAP plots of scRNA sequencing (scRNA-seq) of B-MFs (10,563 cells) and Mo-MFs (10,235 cells) analyzed using Seurat with colors depicting clusters by cell type (**f**) or by gene expression (**g**). **h** Heatmap of top differentially expressed genes (DEG) in 6 major clusters of in vitro-generated B-MF and Mo-MF. **i** Mrc1 expression in B-MF and Mo-MF single cells shown in **f**. **j** scRNA-seq UMAP plot of FACS-purified TAM (10,885 cells) from 4 mice with 4T1.2 cancer cells shows 13 unique cell clusters. Three clusters with overlapping signatures with B-MF are highlighted. **k** Violin plots of three DEG (Egr1, Ier3 and Slc40a1) upregulated in B-MF in vitro and in TAM from mice with 4T1.2 cancer.

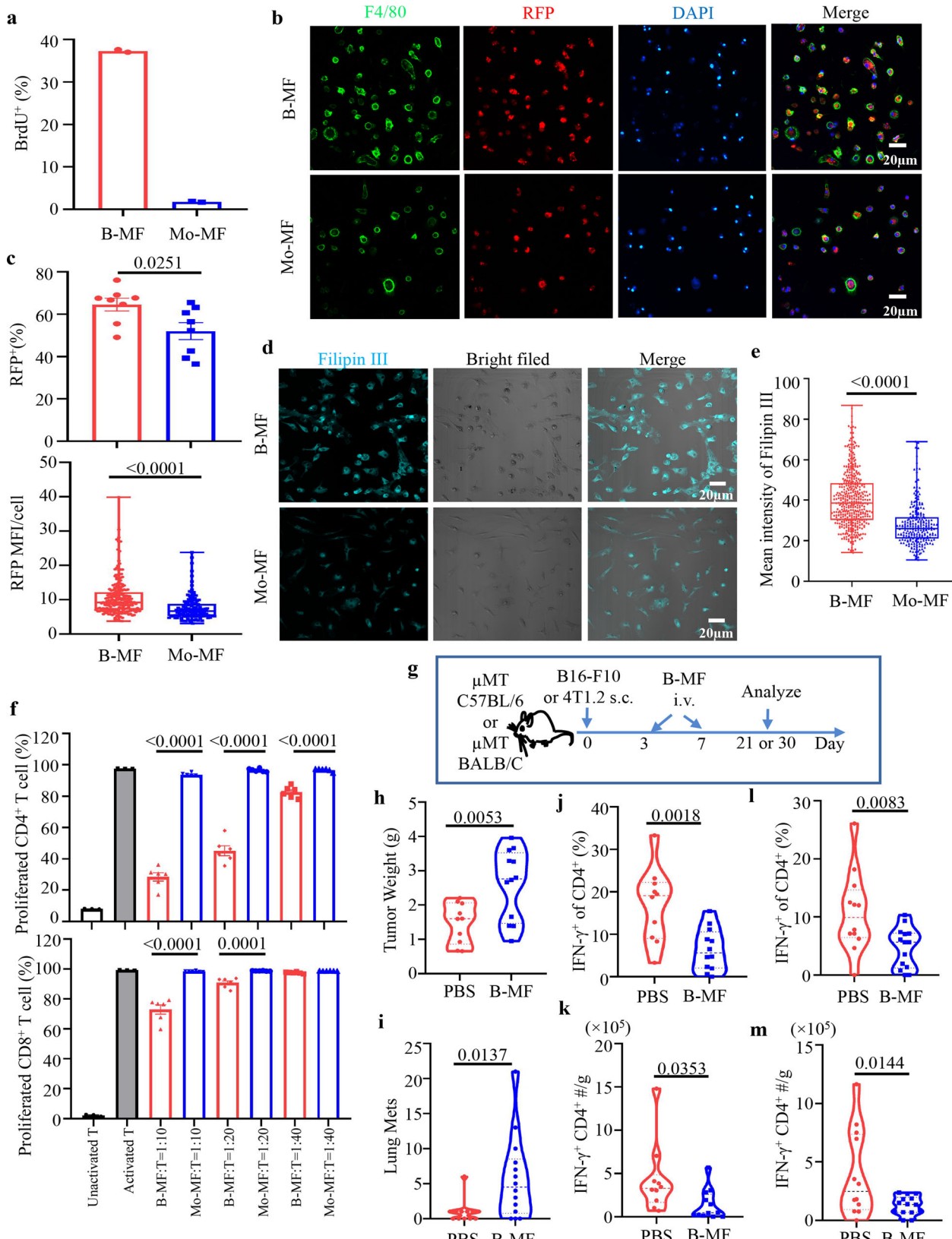

committed lymphoid lineage cells. Despite this, experiments with forced expression or inhibition of a single transcription factor or mutations that drive leukemogenesis[13,14] as well as recent findings of a small proportion biphenotypic CD19+B220+CD16/32++CD11b+ pro-B cells with non-rearranged BCR genes, which become peritoneal CD19+CD79b− macrophages in mice[15], suggest that BMBP retain

plasticity and myeloid transdifferentiation potential. Unlike these artificial manipulations or rare events, here we report that the B-cell-to-macrophage transdifferentiation is commonly used by murine cancers to generate TAM/B-MF. In PB of humans with metastatic/recurrent triple-negative BC and high-grade serous OC, we also detect a significant increase of CSF1R+ CD68+LDLR+ BMBP, which also express the

**Fig. 4 | B-MF and Mo-MF are functionally different. a** Unlike Mo-MF, B-MF incorporate BrdU, i.e., proliferate (BrdU$^+$ frequency ± SEM from 2 mice per group). Compared to Mo-MF, B-MF exhibit higher ability to phagocytize apoptotic ID8-RFP cells than Mo-MF in 2 h assay (**b, c**) and to bind Filipin III (**d, e**). Panels **b, d** show representative fluorescent microscopy images of quantifications of RFP$^+$ cells % ±SEM ($P = 0.0251$) and RFP MFI/cell ($P < 0.0001$) (Mean Fluorescence Intensity MFI, **c**) and Filipin III MFI ($P < 0.0001$, **e**) difference between B-MF and Mo-MF. Eight representative fields per sample were quantified and scale bars represent 20 μm (**c, e**). **f** Unlike Mo-MF, B-MF efficiently suppress proliferation of T cells stimulated with anti-CD3/CD28 Abs for 4 days ($P < 0.0001$ except for CD8 + T cell 40:1 group). $Y$-axis is for Mean proportion ± SEM of CSFE-diluted ($n = 3$ for nonactivated and activated control groups, and $n = 6$ for the rest groups) CD4$^+$ or CD8$^+$T cells when incubated with B-MF or Mo-MF at 10:1, 20:1, and 40:1 ratio ($X$-axis). Control T cells were cultured alone with (activated) or without (nonactivated) anti-CD3/CD28 Abs.

Panels **b–f** were independently reproduced at least three times. **g–m** B-MF support tumor progression. Schema of adoptive transfer experiments in μMT C57BL/6 and BALB/CJ mice with s.c. B16-F10 melanoma and 4T1.2 cancer depicted in **g**. In vitro-generated B-MF ($3 \times 10^5$) from C57BL/6 and BALB/c mice were i.v. transferred into μMT C57BL/6 and μMT BALB/c mice, respectively, at days 3 and 7 post-tumor challenge. Shown are quantifications of tumor weight in mice with B16-F10 melanoma ($n = 10$ for PBS and $n = 12$ for B-MF, $P = 0.0053$, **h**), metastatic foci in the lungs of mice with 4T1.2 cancer ($n = 12$ for PBS and $n = 14$ for B-MF, $P = 0.0137$, **i**), and frequency and absolute numbers of IFNγ$^+$CD4$^+$ T cells per gram primary tumor in mice with B16-F10 melanoma (**j**, $P = 0.0018$ and **k**, $P = 0.0353$) and 4T1.2 cancer (**l**, $P = 0.0083$ and **m**, $P = 0.0144$). $P$-values in **c, e, f, h–m** was calculated using two-tailed unpaired $t$-test. Results were independently confirmed at least twice. Each symbol in **h–m** is for a single mouse.

macrophage-specific genes *Cebpa*, *Cebpb*, and *Marco*. Importantly, the B-MF signature is also identifiable within unique macrophage clusters using recently published scRNA-seq profiles of tumor-infiltrating cells in patients with BC[38] and high-grade serous OC[39]. Our results suggest that human and murine cancers primarily transdifferentiate BMBP into macrophages, adding one more feature to the heterogeneity and plasticity of TAM. The inflammatory and antitumor activities of TAM at the early stages of the tumor can shift to proangiogenic and tumor-supporting M2-like phenotypes as the tumor progresses[40], presumably when B-MF would be induced. Interestingly, B-MF resembles both small (S)-TAM and large (L)-TAM (which is associated with a poor disease outcome) recently identified in human colorectal liver metastasis[41]. The lipid metabolism and phagocytosis genes of B-MF (*Fasn*, *Pltp*, *Acat1*, *C1qa*, and *C1qb*) are upregulated in L-TAM, while *LDLR*, *Hmgcr* as well *S100a8*, *Vcan*, and *Thbs1* are increased in S-TAM.

Although trogocytosis or cell fusion can be mistaken as transdifferentiation[27,28], we show that it only accounts for a very minor fraction of B-MF. Instead, the overwhelming majority of B-MF derived from highly FACS-purified CD45.2$^+$ B-cell precursors of the B-cell lineage tracer Mb1-EYFP$^+$ mice only expressed CD45.2 after in vitro co-differentiation with CD45.1$^+$ monocyte/macrophages or adoptive transfer in tumor-bearing CD45.1$^+$ mice. Our results show that at least some TAM originate from bona fide B cells besides their hitherto source, monocytes[10]. The biological relevance of this redundancy in the generation of TAM remains poorly understood; however, based on our comparisons of the side-by-side generated B-MF and Mo-MF, we think that the two macrophages may serve different purposes. Transcriptionally, B-MF preferentially upregulate the expression of genes involved in the cell cycle, fatty acid metabolism, and steroid-cholesterol biosynthesis, implying they utilize unique metabolic and inflammatory functions. Unlike Mo-MF, B-MF proliferate, i.e., self-maintain, and thus may persist longer in the tumor. B-MF markedly upregulate surface expression of LDLR, which removes extracellular cholesterol/LDL[42], and this could explain the higher levels of intracellular cholesterol and lipids in B-MF compared to Mo-MF. Consistent with significant upregulation of genes associated with phagocytosis, M2-skewing and immunosuppressive functions (*PD-L2*, *B7-H3*, *Marco*, *TGFβ*) and downregulation of pro-inflammatory and IFNγ response genes, B-MF express higher levels of surface MRC1 (CD206), PD-L1 (CD274), and TGFβ/LAP and efficiently phagocytize apoptotic cells compared to Mo-MF. This efficient phagocytosis presumably occurs without overt inflammation, as LDLR-mediated cholesterol influx inhibits activation of the inflammasome[43]. Our data show that cancer generates phenotypically and functionally nonredundant TAM from BMBP and monocytes, where B-MF appear to promote cancer growth presumably by controlling antitumor T-cell responses. First, unlike Mo-MF, B-MF efficiently suppress the proliferation of T cells or induce the generation FoxP3$^+$ Tregs in vitro. Second, B-MF significantly increase the growth of B16-F10 melanoma and lung metastasis of 4T1.2 breast cancer in two different strains of μMT mice. To do this, they primarily

decrease tumor-retarding IFNγ$^+$ CD4$^+$ T cells in the tumor[31], presumably by utilizing the B-MF-expressed immunoregulatory factors, TGFβ/LAP and PD-L1[25,26], and LDLR. For example, LDLR may enhance the TGFβ responsiveness of target T cells by removing extracellular LDL/cholesterol that impairs TGFβ binding and thus signaling via TGFβRII/TGFβR1[44].

We propose that cancer primarily targets a small subset of CSF1R$^+$Pax5$^{Low}$ pre-B cells and iB cells recently emigrated from BM. First, B-MF are not found in tumor-bearing mice with B-cell differentiation blockage at the pro-B-cell stage. Second, splenic transitional B cells from naïve mice do not generate B-MF regardless of their CSF1R expression state, as cancer first needs to mobilize BMBP into circulation as the source of B-MF. We and others have reported that cancers use TSLP and G-CSF to mobilize BM pre-B cells and HSPS in circulation[11,45]. We also find the chromatin accessibility landscape of BM CSF1R$^+$ BMBP to be significantly more open and permissive to macrophage-differentiation signals. In contrast, CSF1R$^+$ and CSF1R$^-$ splenic B cells and BM CSF1R$^-$ cells present a chromatin landscape that is refractory to macrophage conversion. The overwhelming majority of regions with decreased accessibility in the BM CSF1R$^-$ cells remained closed in the spleen (679 of 749), presumably "locking in" the state of lymphoid lineage potential. It appears that BM CSF1R$^+$ cells have a potentially more primitive, permissive chromatin state allowing for myeloid lineage transformation. The BM CSF1R$^+$ cells have more accessible ERG and RUNX1 binding sites, two transcription factors expressed in myeloid and lymphoid progenitor cells[34] and early progenitor and myeloid-committed progenitor cells[37], respectively. Given that RUNX1 also upregulates the expression of *Csf1r* by binding to its promoter and enhancer regions[36] and that the PAX5 deletion alone removes the repression of *Csf1r* and other myeloid lineage-specific genes and induces the BM B-cell precursor transdifferentiation[14,33], we think that RUNX1 supports CSF1R expression in CSF1R$^+$ BMBP to downregulate Pax5 in response tonic and cancer-secreted M-CSF. Given that the CSF1R$^+$ BMBP in our patients with metastatic/recurrent triple-negative BC and high-grade serous OC co-express macrophage-associated genes, human cancer may also target biphenotypic B-cell precursors reported to have a macrophage-differentiation potential[15]. Overall, our data indicate that the B-cell-to-macrophage transdifferentiation is a physiological and widely utilized phenomenon. Murine and possibly human cancers target the transdifferentiation to generate immunosuppressive TAM.

## Methods

### Mice and cell lines

The animal protocol was approved by the ACUC committee of the National Institute on Aging (ASP 322-LMBI-2022) under the *Guide for the Care and Use of Laboratory Animals* (NIH Publication No. 86-23, 1985). The study used young (8–12 weeks old) female mice bred and housed in the same, specific pathogen-free environment at the National Institute on Aging (NIA). C57BL/6 J, BALB/CJ, R26R-EYFP

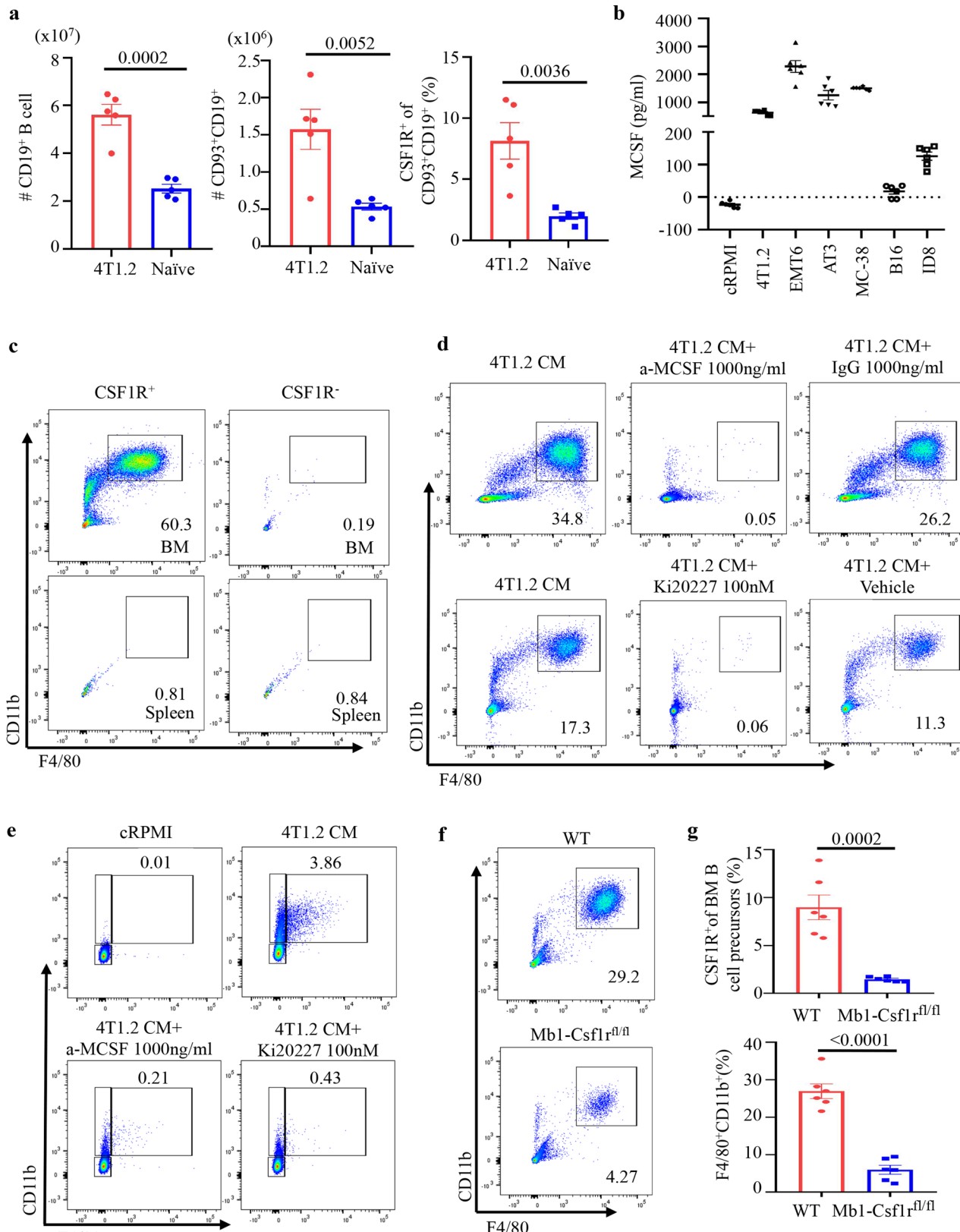

(B6.129×1-Gt (ROSA)26Sortm1(EYFP)Cos/J), Csf1r^flox mice (B6.Cg-Csf1rtm1.2Jwp/J) and μMT mice (B6.129-Ighm-tm1Cgn/J) and J_HT mice (J_HT; B6.129P2-Igh-Jtm1Cgn/J) in C57BL/6 background mice were purchased from the Jackson Laboratory (Bar Harbor, ME); RAG2-GFP mice expressing bacterial artificial chromosome modified GFP instead of RAG2 were a gift of Dr. Michael Nussenzweig (Howard Hufhes Medical

Institute, NY, NY) and reported elsewhere[46], μMT mice in BALB/c background were a gift from Dr. Thomas Blankenstein (Max-Delbrück-Center for Molecular Medicine, Berlin, Germany)[47]. Mb1-Cre mice in C57BL/6 background (B6.C(Cg)-Cd79atm1(cre) Reth/EhobJ) were a gift from Dr. Richard Maraia (National Institute of Child Health and Human Development, Bethesda, MD)[48]. Mogp-tag mice (Mogp, spontaneous

**Fig. 5 | Cancer mobilizes CSF1R⁺ BMBP into the circulation to generate B-MF via signaling CSF1/CSF1R axis. a** Numbers ± SEM of CD19⁺ ($P = 0.0002$, left) and CD93⁺CD19⁺ ($P = 0.0052$, middle) B cells and frequency ± SEM of CSF1R⁺ within CD93⁺CD19⁺ B cells in the spleen of naïve vs 4T1.2 cancer-bearing BALB/c mice ($P = 0.0036$, right, $n = 5$). **b** ELISA measurements of secreted M-CSF in CM by 4T1.2, EMT6, AT3, MC38, B16-F10, and ID8 cells ($n = 6$, pg/ml). **c–e** Representative FACS plots of the F4/80⁺CD11b⁺ B-MF converted from highly FACS-purified BM CSF1R⁺ and CSF1R⁻ B-cell precursors and splenic CSF1R⁺ and CSF1R⁻ B cells (**c**), BMBP after 7-day (**d**), or pre-B-cell line 70z/3 after 30-day (**e**) culture in 4T1.2-CM alone or in the presence of neutralizing anti-M-CSF Ab or a specific CSF1R inhibitor Ki20227 (**d**, **e**). **f, g** Representative FACS plot (**f**) and quantification (frequency ± SEM, **g**) of CSF1R⁺ BM B cells in Mb1-CSF1R^Flox/Flox mice as compared to WT littermates ($P = 0.0002$, top panel) used for the generation of B-MF after 7-day 4T1.2-CM treatment ($P < 0.0001$, lower panel, $n = 6$ mice). Gating is shown in Supplementary Fig. 8. In **a**, **b**, and **g**, each symbol is for a single mouse. $P$-values in **a**, **b**, and **g** were calculated using two-tailed unpaired $t$-test. Results for all panels were independently reproduced at least three times.

ovarian cancer model in C57BL/6 mice) were a gift from professor Dr. I. Miyoshi (Tohoku University Graduate School of Medicine, Miyagi, Japan)[6]. To create mice with B-cell-specific EYFP reporter (Mb1-EYFP) or CSF1R deletion (Mb1-CSF1R^Flox/Flox), Mb1-cre mice were bred with R26R-EYFP and Csf1r^flox mice, respectively.

4T1.2 cells were a gift from Dr. Robin L. Anderson (Peter McCallum Cancer Center, Melbourne, Australia); MC38 colonic adenocarcinoma cells were a gift from Dr. Jeffrey Schlom (National Cancer Institute, Bethesda, MD)[49]; mammary carcinoma AT3 cells were a gift from professor Scott I. Abrams (Roswell Park Comprehensive Cancer Center, Buffalo, NY); ID8-p53⁻/⁻-RFP (ID8 or ID8-RFP) cells were a gift from professor Sharon Stack (University of Notre Dame, IN); and EMT6 cells and melanoma B16-F10 cells were purchased from American Type Culture Collection (Manassas, VA). Cells were tested free of mycoplasma with Mycoplasma Detection Kits (Lonza Basel, Switzerland; and IDEXX BioAnalytics, Columbia, MO).

### Tissues and blood processing
PBMC from healthy human donors were collected with written informed consent at the Clinical Core Laboratory, NIA, under Human Subject Protocol # 2003054 and Tissue Procurement Protocol # 2003-071; and from patients with recurrent breast and ovarian cancer[50,51] enrolled in Phase II clinical study of prexasertib (NCT02203513) at the Clinical Center, Center for Cancer Research, National Cancer Institute. All patients, including 13 participants in this research project, provided written informed consent before enrolment and on using their samples for research. The study has been conducted in accordance with ethical principles that have their origin in the Declaration of Helsinki and are consistent with the International Council on Harmonization guidelines on Good Clinical Practice, all applicable laws and regulatory requirements, and all conditions required by a regulatory authority and/or institutional review board. The study protocol was approved by the Institutional Review Board of the Center for Cancer Research, National Cancer Institute. All experiments were performed on PBMC, which were cryopreserved after collection. Mouse BM cells were flushed out of femurs and tibias with cold cRPMI. Single-cell suspension of BM, spleen, LN was prepared with 70 μm strainer (Falcon, Bedford, MA). BM, spleen, and blood cells were treated with ACK buffer to remove red blood cells. Mouse tumor tissues were cut into 3–5 mm pieces and digested with a mouse tumor dissociation kit (Miltenyi Biotec, Bergisch Gladbach, Germany) following the manufacturer's instructions.

### Flow cytometry (FACS)
For immune cell phenotyping, cells were pre-incubated with TruStain FcX™ solution before immunostaining with different combinations of anti-mouse or anti-human Abs (1 μg per 10⁶ cells, Supplementary Table 1) and fixable viability dye, then fixed/permeabilized with eBioscience™ intracellular fixation & permeabilization buffer (Thermo Fisher, Waltham, MA). The samples were evaluated on FACSymphony™ (BD, Franklin Lakes, NJ), Amnis ImageStreamX MKII (Millipore, Burlington, MA), or CytoFLEX (Beckman Coulter, Brea, CA). The results were analyzed with FlowJo v10(BD), IDEAS (Millipore), or Cytoexpert 2.3 (Beckman).

### Immunofluorescent staining
Dissected tumors from mice were fixed with 4% PFA in PBS for 24 h and then transferred to 30% sucrose in PBS for about 2 days until the tissue sank to the bottom of 15 ml Falcon tubes. Tumors were embedded in OCT compound, frozen on dry ice, and stored at −80 °C before cryosection. Ten micrometer thick sections were prepared and adhered to superfrost glass slides. After three washes with PBS, the tumor slices were incubated in 0.3 M glycine in PBS for 30 min and then blocked and permeabilized with IF buffer (5% donkey serum, 2% BSA, and 0.1% Triton X-100 in PBS) for 60 min at room temperature (RT). Tumors slices were incubated with anti-CD19 (abcam, Cat # ab245235, dilution 1:100, final concentration 4.6 μg/ml) and anti-CD68 (abcam, Cat # ab53444, dilution 1:300, final concentration 3.3 μg/ml) antibodies for 24 h at 4 °C. After three washes with PBS, the slices were incubated with donkey anti-rabbit IgG H&L Alexa Fluor 488 (Abcam, Cat # ab150073, dilution 1:500, final concentration 4 μg/ml) and donkey anti-rat IgG H&L Alexa Fluor 568 (Abcam, Cat # ab175475, dilution 1:500, final concentration 4 μg/ml) at RT for 2 h. After washing with PBS three times, slides were mounted with ProLong™ Diamond Antifade Mountant with DAPI (Invitrogen) and imaged using a Zeiss LSM 710 confocal microscope.

### Cancer CM media preparation and cytokine quantification
Cells were cultured in RPMI1640 or DMEM (for ID8 cells) supplemented with 10% FBS, 1× HEPES, sodium pyruvate, nonessential amino acids solution, penicillin−streptomycin−glutamine (Gibco, Gaithersburg, MD), and 55 mmol/L β-mercaptethanol in T75 flask to 70−80% confluency. CM was collected after 5 min centrifugation at 1500 rpm, filtered with 0.2 μm filter, and stored at −80 °C as single-use aliquots. For cytokine tests, confluent cells were cultured with RPMI without FBS for 24 h. Mouse serum was collected using BD Microtainer® Tubes following the manufacturer's instruction. Cytokines and M-CSF in filtered CM or sera were evaluated with Quantikine ELISA kit (R&D, Minneapolis, MN) or with Proteome Profiler Mouse XL Cytokine Array (R&D). Images were captured and analyzed with Fiji software.

### B-MF conversion assay
BM Lin⁻ (TER119, CD11b, Gr-1, CD3ε, NK1.1 or CD49b, Ly6C, Ly6G, CD11c)⁻ CD19⁺ B cells were isolated from C57BL/CJ or BALB/CJ mice using FACSAria™ Fusion sorter and 10⁶/ml B cells were cultured in 50% cancer CM in cRPMI for 7 days in Nunc™ Multidishes with UpCell™ Surface (Thermo Fisher) without changing media for 7 days. 70z/3 pre-B cells (10⁵/ml) were cultured in 50% cancer CM for up to 30 days with a replenishing culture medium every 3−4 days. Adherent cells (macrophages) were harvested by detaching them at 4 °C for 15 min in PBS. For Giemsa staining, B-MF was fixed with ethanol for 5 min, and Wright-Giemsa stained according to the manufacturer's instructions. CSF1R receptor signaling was blocked with Ki20227 (R&D).

### In vitro assays
For bacterial uptake assay, E. coli (Thermo Fisher) labeled with pHrodo™ red (0.1 mg/ml) were cultured with B-MF generated from RAG2-GFP for 2 h. Cells were washed with PBS, fixed with 4% formaldehyde, and stained with DAPI. For phagocytosis of apoptotic cancer cells, ID8-RFP cells (10⁶/ml) were pretreated with 300 nM

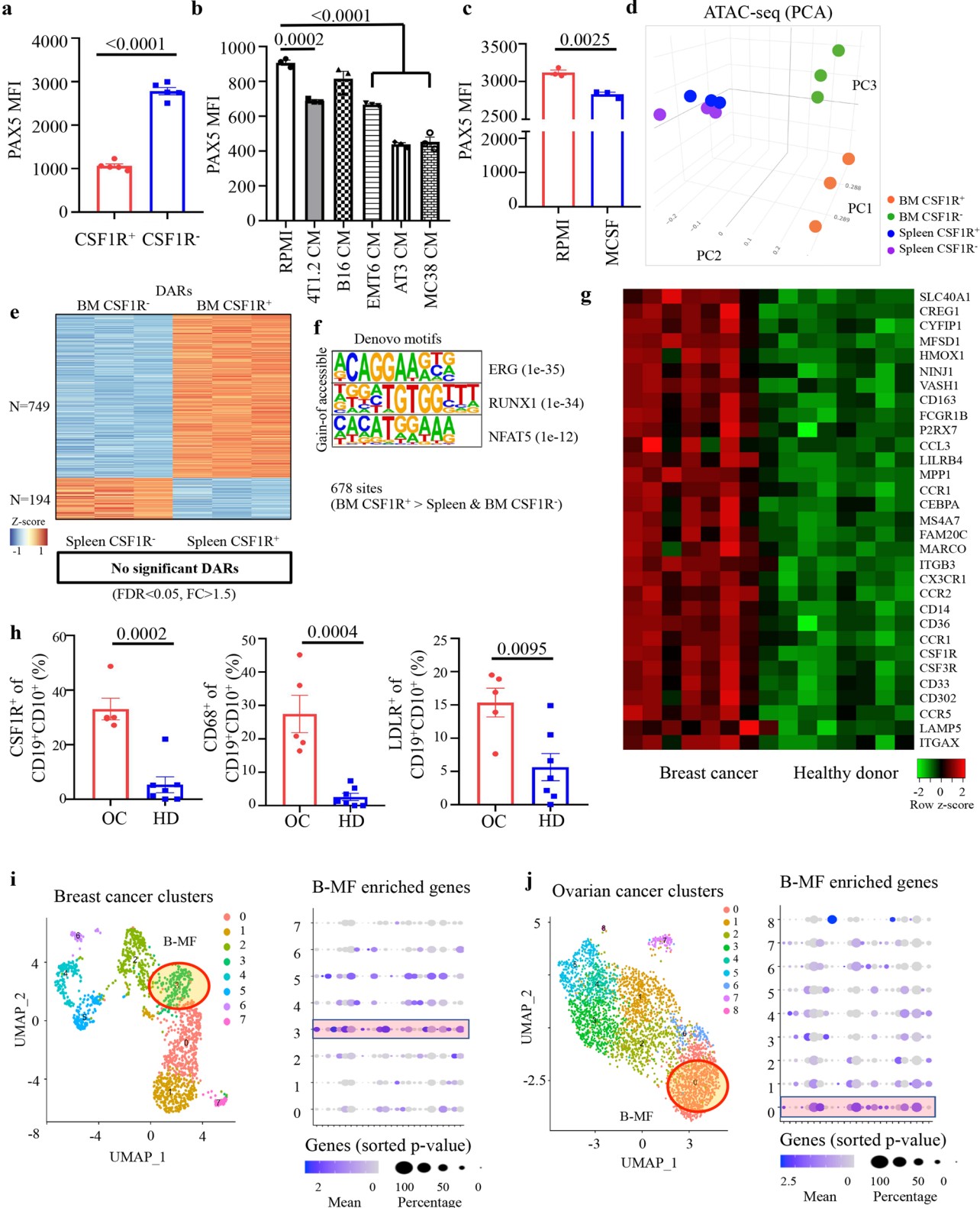

gemcitabine hydrochloride (Sigma, St. Louis, MO) for 24 h, then washed with PBS and cultured with macrophages for 2 h. Macrophages were stained with anti-F4/80-FITC Ab, and DAPI and phagocytosis was evaluated using Zeiss LSM 710 (Carl Zeiss AG, Jena, Germany) and analyzed with Fiji software. For macrophage proliferation test, BrdU (10 μM, BD) was added to macrophage cultures on day 5, and the BrdU incorporation was quantified on day 7 using FACSymphony™ and analyzed by FlowJo.

T-cell suppression assay was described elsewhere[12]. Briefly, splenic T cells isolated with CD3+ T-cell enrichment column (R&D) were labeled with eFluor™ 450 and cultured with macrophages at 1:10, 1:20, and 1:40 E:T ratios in 96-well flat-bottom plates coated with 5 μg/ml anti-mouse CD3e antibody (clone 145-2C11, BD) and free anti-mouse CD28 antibody (2 μg/ml, clone 37.51, BD) for 4 days. The Treg conversion assay was described elsewhere[5]. In brief, FACS-sorted splenic CD4+CD25− T cells were cultured with macrophages at 1:5, 1:10, and

**Fig. 6 | Cancer targets BM CSF1R⁺PAX5^Low B-cell precursors. a−c** Pax5 MFI (Mean ± SEM) in freshly isolated BM CSF1R⁻ vs CSF1R⁺ B-cell precursors ($P < 0.0001$, $n = 5$ mice, **a**); in pre-B 70z/3 cells treated with indicated cancer CM ($P = 0.0002$ R*PMI* vs 4T1.2-CM, $P < 0.0001$ R*PMI* vs EMT6 CM, AT3 CM and MC38 CM, $n = 3$ independent cell cultures, **b**); and BM Lin⁻ CSF1R⁺ CD19⁺B220⁺CD93⁺IgM⁻IgD⁻ B-cell precursors after treatment*RPMI* vs M-CSF for 48 h ($P = 0.0025$, $n = 3$, **c**). *P*-values in **a**−**c** were calculated using two-tailed unpaired *t*-test. **d** 3D PCA plot of chromatin accessibility data of BMBP CSF1R⁺ and CSF1R⁻ from BM and spleen of naïve mice ($n = 3$ per group). **e** Heatmap of differentially accessible regions (DARs) in BM CSF1R⁺ and CSF1R⁻ BMBP. No significant DARs with FDR < 0.05 and FC (fold change) >1.5 were detected in splenic cells. **f** Significant de novo motifs predicted from 678 sites that are more open in BM CSF1R⁺ compared to both BM and splenic CSF1R⁻ BMBP. **g** mRNA microarray heatmap of macrophage-related DEGs in B cells isolated from PB of patients with breast cancer (BC, $n = 8$) compared to healthy donors (HD, $n = 7$). Scale bar is for expression z-score. **h** Frequency ± SEM of CSF1R⁺ ($P = 0.0002$, left), CD68⁺ ($P = 0.0004$, middle), and LDLR⁺ ($P = 0.0095$, right) cells within CD19⁺CD10⁺ B cells from PB of patients with ovarian cancer (OC) vs healthy donors (HD) ($n = 5$ for OC, $n = 7$ for HD). *P*-values in **h** were calculated using two-tailed unpaired *t*-test. **i, j** UMAP of scRNA-seq data of macrophages (left) and expression levels of the in vitro-generated B-MF genes (right) in published human BC (**i**) and OC (**j**) datasets. Highlighted regions show clusters with overlapping expression signatures of B-MF.

1:20 E:T ratios in plates coated with 5 µg/ml anti-mouse CD3e antibody and free recombinant murine IL-2 (5 ng/ml, PeproTech, Rocky Hill, NJ) in for 5 days. Control T cells were cultured with recombinant mouse TGF-β1 (5 ng/ml, R&D) in cRPMI without macrophages.

## In vivo experiments

For evaluation of macrophages in vivo, tumor cells were subcutaneously injected into congenic mice, such as 4T1.2 cells and EMT6 cells ($1 \times 10^6$) in BALB/cJ and µMT mice, and B16-F10, and AT3 and MC38 cells ($1 \times 10^6$) in C57BL/6 J, J_HT or Mb1-EYFP mice. ID8-p53⁻/⁻-RFP cells ($5 \times 10^6$) were i.p. injected into C57BL/6 J mice. Spontaneous Mogp cancer grows in the peritoneum of C57BL/6⁶. For in vivo B-MF generation study, C57BL/6 J mice with ID8-p53⁻/⁻-RFP cells ($5 \times 10^6$) in PeC were i.p. injected with BM Lin⁻CD19⁺EYFP⁺ B cells ($5 \times 10^6$) sort-purified from Mb1-cre-EYFP mice and then 7 days later, the PeC lavage cells were FACS evaluated. To evaluate the tumor-supporting role of B-MF in vivo, µMT mice were intravenously injected with $3 \times 10^5$ in vitro-generated B-MF or PBS 3 and 7 days after subcutaneous challenge with B16-F10 melanoma cells (day 0) or 4T1.2 breast cancer cells ($0.5 \times 10^5$) in the fourth mammary gland and the lungs were analyzed as we previously described[5,6]. Tumor volume (V = a × b, mm²) for B16 melanoma was measured on days 11, 14, 16, 18, and 21, and on day 21, mice were euthanized to evaluate tumor weight and T cells. For B-MF tracking experiment, in vitro-generated B-MF from BALB/c mice were eFluor™ 450 labeled and i.v. injected ($5 \times 10^5$ cells/mouse) into µMT BALB/c mice with 14-day s.c. 4T1.2 tumor. To compare B-MF to B cells, µMT BALB/c mice were i.v. injected with naïve BALB/c mouse in vitro-generated B-MF or FACS-purified FOB ($3 \times 10^5$ cells/mouse) 3 and 7 days after s.c. challenge with 4T1.2 cells. Lung mets and TILs were quantified at day 30 post-tumor challenge.

## Cellular cholesterol content quantification

Macrophages were fixed with 4% formaldehyde solution in TBS for 5 min, then after TBS washes, they were incubated with Filipin III at 1:100 dilution in TBS (5 mg/ml stock in 100% ethanol, Cayman, Ann Arbor, MI) for 60 mins in the dark. Cells were washed with TBS, and lipids were quantified with Zeiss LSM 710 and Fiji software, as described above.

## mRNA microarray

For the collected biological samples, the standard RNA extraction protocol was performed by RNeasy Plus Micro kits (QIAGEN, Hilden, Germany), and genome-wide expression was measured using the Agilent platform (Mouse 8X60K v2 and Hs 8X60K v3, Agilent, Santa Clara, CA, USA) according to the manufacturer's instruction. Principal Component Analysis (PCA) was performed using the Prcomp R function with expression values. Differentially expressed genes (DEGs) were assessed using the moderated (empirical Bayesian) *t*-test implemented in the limma package (version 3.14.4)[52], and correction for multiple hypothesis testing was accomplished by calculating the Benjamini−Hochberg false discovery rate. Enriched pathways were discovered by GSEA tool[53] with Molecular Signature Database v7.4. All microarray analyses were performed using the R environment for statistical computing (version 3.6.2).

## scRNA-seq

Sort-purified single-cell suspensions were loaded into a 10× Chromium controller (10x Genomics, Pleasanton, CA, USA) and converted to a barcoded single-cell RNA expression library according to the standard protocol of the Chromium Next GEM Single cell 3′ kit (v3.1 chemistry) in Laboratory of Immunology and Molecular Biology, National Institute on Aging, and the single-cell 3′ gene expression libraries were sequenced on NovaSeq 6000 (Illumina, San Diego, CA, USA) in the Genomics Core facility of the Johns Hopkins School of Medicine. Raw sequencing data were processed using the Cell Ranger version 5.0 (10x Genomics, Pleasanton, CA, USA) pipeline. The raw gene expression matrix was normalized and scaled using the SCTransform method[54] in the Seurat R package (version 4.0)[55]. The minimum number of detected genes was set to 1000, and genes were chosen when they were detected in more than three cells. Dimension reduction was performed using principal component analysis (PCA). For visualizing the generated clustering, we used the Uniform Manifold Approximation and Projections (UMAP) plot. We defined clusters with a leiden algorithm using shared nearest neighbor (SNN) in PCA space. From in vitro B-MF and Mo-MF, we generated a total of 12 clusters for in vitro samples (0−11). Integration of in vivo samples with canonical correlation analysis (CCA) was performed, and we generated 13 clusters for in vivo tumor macrophage samples (0−12). Finally, we performed a nonparametric Wilcoxon rank-sum test to search for highly expressed genes in the clusters. In addition, human tumor single-cell transcriptomes were downloaded from GEO (GSE114725, and GSE146026) and also processed with the same pipeline described. We used only macrophage clusters for downstream analysis. All single-cell analyses were performed using the R environment for statistical computing (version 4.0.5).

## ATAC-seq

We utilized a Hi-Seq 2000 machine to sequence the ATAC-seq libraries (Illumina, San Diego, CA). We prepared 12 pair-end ATAC-seq libraries including BM CSF1R (±) and Spleen CSF1R (±) samples ($n = 3$ per group). In total, 369 M reads were sequenced, and average 31 M reads were sequenced per sample. We applied NIEHS TaRGETII ATAC-seq pipelines, which are available to the genomics community. All raw reads were trimmed using cutadapt package, and trimmed reads (>36 bp minimum alignment length) were mapped against the mm10 reference genome using BWA aligner[56]. We used de-duplicated and uniquely mapped reads for peak calling analysis after excluding black-list regions defined by ENCODE[57]. The candidate peaks were predicted by MACS peak calling tool[58]. In addition, we also applied the DESeq2[59] to determine differentially accessible regions (DARs); cutoff: Fold change > 1.5, log2CPM > 1.2, FDR < 0.05. The differentially accessible regions were submitted for the search of potential transcription factor binding sites using HOMER software[60]. We used non-DARs as background regions in de novo motif analysis.

## Statistical analysis

The results are presented as the mean with each individual data point or in bar graph ± SEM. GraphPad Prism (Prism 6; GraphPad Software, Inc) was used to perform statistical analysis. Data were analyzed using Welch *t*-test or one-way ANOVA. A *P*-value less than 0.05 was considered significant (****$P < 0.0001$, ***$P < 0.001$, **$P < 0.01$, and *$P < 0.05$).

## Reporting summary

Further information on research design is available in the Nature Research Reporting Summary linked to this article.

## Data availability

The authors declare that the data that support the findings of this study are available within the Article and its Supplementary Information file. RNA-seq and ATAC-seq data are deposited in https://www.ncbi.nlm.nih.gov/geo/query/acc.cgi?acc=GSE178716 and https://www.ncbi.nlm.nih.gov/geo/query/acc.cgi?acc=GSE180285. Source data are provided with this paper.

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

## Acknowledgements

We are grateful to Mrs. Jane Trepel with help with human BC and OC samples; Drs. Elin Lehrman, Yongqing Zhang, and Supriyo De (Genomics Core, LGG, NIA) for help with performing microarray experiments and initial data analyses, and data submission to GSE. This research was supported by the Intramural Research Program of the National Institute on Aging and in part, the National Cancer Institute, NIH.

## Author contributions

C.C. performed the research, collected, and analyzed data; E.R., M.B., X.W., and L.Z. performed experiments; B.P. worked on bioinformatics analyses, J.M.L. provided clinical trial samples; I.B. supervised bioinformatics analysis; A.B. conceived, designed, and supervised the study and wrote the manuscript.

## Funding

## Competing interests

The authors declare no competing interests.
