## [Peer Review File · Nature Communications]

Cancer co-opts differentiation of B-cell precursors into macrophage-like cellsREVIEWER COMMENTS

Reviewer #1 (Remarks to the Author):

The paper by Chen et al investigates the effect of the tumor environment on immature B-cells. Using tumor models the authors provide evidence for that the B-cells respond to M-CSF and potentially other factors in the tumor microenvironment by changes in the transcriptional program to express Macrophage markers and to gain both phagocytic and immune suppressive abilities. The authors re-analyze single cell gene expression data from primary human tumors indicating the presence of atypic cells resembling the tumor converted B/MF-cells observed in the mouse model. The authors also provide some evidence for that the presence of these B/MF cells allows for more extensive tumor growth.

While this is an interesting paper using a wide variety of techniques and approaches. It is difficult to determine how unique this population is for a tumor microenvironment and how important the population is in the suppression of tumor rejection. There are certain controls missing making portions of the data difficult to put into context and thereby do the correct interpretations of the results. Please see specific comments below.

Specific comments:

1: One of my major concerns relates to how unique this population is for the tumor environment. The B-MF cells appear rare among the TIB, are these cells found in spleen or BM of non-tumor inoculated mice. Are they present outside of the tumor environment in the tumor inoculated mice? Analysis of the formation of these cells using a lineage tracer are done in panel 1I but then using a new tumor model as compared to 1A and S1C. In 1I mice without tumor obtain over 2% F480CD11b cells, higher levels than what is reported among TIB in Figure 1A. There are experiments to this end presented in Figure S2. Here, however, it is unclear what "naïve" stand for. Wt mice? As the text is written it reads as if naïve would be mice not exposed to MC38. "EYFP+ F4/80+CD11binter/low TAM were significantly increased in the tumor (Fig. S2)." . If so, what organ was used for analysis of control mice, where is the statistical analysis? and why are not the CD19+ cells in the control mice YFP+? This needs to be resolved. I would strongly recommend that the presence of B-MF cells in normal as well as tumor inoculated mice is carefully investigated to determine if this indeed represent a tumor specific population and if so, is it restricted to the tumor environment?

2: A second concern relates to the actual cell of origin. The data suggest that the cells are generated from progenitor cells. However, as the absolute majority are IgM+ in the absence of surrogate light chain expression (Figure S4). This strongly suggest that the cell of origin is an Immature B rather than IgM negative progenitors. Are BM progenitors from μ MT mice converted in the in vitro culture systems. It is also unclear to me how frequent CD19+ cells are in the tumor environment in the μ MT mice. If there are a substantial number of CD19+ cells in these mice they should all be progenitors and if not converted, would this not be conclusive evidence for that this is a unique ability of immature B-cells?

3: The authors claim "4T1.2-CM, but not control medium, significantly upregulated F4/80 and CD11b in B cells from BM, but not from spleen, LN or PeC (Fig. 1C)." However, even though such data are shown for 70Z3 cells in panel 1F and G after 30 days of cultivation (The 30 day cultivation time should be indicated in the text), I cannot understand that Figure 1C contains data on either control medium or any statistical analysis of data from primary cells. This needs to be clarified and corrected.

4: Figure 1D as well as S3B lack pictures of cells cultivated in "control media".

5: One of the most interesting findings is that immune suppressive function of the B-MF cells. This is studied in one experiment, using one tumor model and without any detailed analysis of the tumor environment. It would be important to know if the 300 000 B-MF transplanted home to the tumor. The ability of these cells to proliferate could potentially cause an additional expansion of this population in μ MT mice. How much of these cells are found in the tumor at the time of analysis? Are these levels comparable to what is observed in the tumor microenvironment in Wt

mice? As the B-MF cells are CD45+, could that result in an apparent decrease in the T-cell fraction? Absolute number would be more relevant. As this is the most important part of this paper, I would recommend that this analysis is extended to involve more tumors as well as a much more detailed analysis of the tumor environment.

Minor comments:

1: I do not agree with the title of the paper as the cells generated do not appear to be conventional MF as they obtain unique transcriptional profile.

2: While it is clear from the data that the BM B-cells responds by upregulation of MF genes and a down regulation of B-lineage genes, it is rather unexpected that they to a large extent retain expression of CD79a (a gene by the way missing in the heat map in figure S4C) and IgM. As CD79 is a PAX5 target this would need to be discussed.

3: What are the numbers in the FACS plots representing, average? representative exp? If average why not std inserted.

4: I would be useful to stress that this is a lineage tracer hence it does not report CD79 alpha expression but rather a history of CD79a expression where this model first is referred to.

5: Gene list in 2F appears misaligned.

Reviewer #2 (Remarks to the Author):

In this manuscript Chen et al, describe that cancer cells use M-CSF to reprogram a rare subset of M-CSFR+ B cell progenitors into macrophages with putative pro-tumorigenic functions. This study extends previous observations from other groups . Unfortunately, the data presented does not rigorously support the central premise of the manuscript. Additionally, key controls are missing and results are inconsistent across figure panels. In many cases it is very difficult to determine which cells are being analyzed and how. Below are the main limitations of the manuscript and my suggestions on how to improve it.

Demonstration of transdifferentiation: the central claim of the manuscript is that cancer reprograms a rare subset of B cell precursors into macrophages. This claim must be rigorously supported. Studies claiming transdifferentiation of hematopoietic cells were later shown to be due to cell fusion when more rigorous and well-controlled experiments were performed (e.g. Science. 2002;297:1299, Nature. 2003 Apr 24;422(6934):897-901, Nat Med. 2004 May;10(5):494-501). Further, cell fusion is enhanced by inflammation (Nat Cell Biol. 2008 May;10(5):584-92, Nat Cell Biol. 2008 May;10(5):575-83). The authors data supporting that B cell precursors give rise to macrophages is based on the identification of CD11b+F4/80+ cells within the CD19+ gate, that cultured B cell precursors generate macrophages in vitro, that B cell macrophages are lineage traced after transfer of B cell precursors from the Mb1-cre:EYFP mice, and that Mb1-cre:Csf1rflox/flox mice have impaired capacity to produce B macrophages. There are significant limitations and missing controls with these experiments.

a) The authors should demonstrate that the B cell macrophages are generated by transdifferentiation and not via cell fusion. The easiest way to examine this will be to transfer Mb1-YFP+ precursors into Ubc-gfp (or other pan fluorescent protein reporter) mice and demonstrate that the B cell macrophages are GFP-.

b) The authors culture FACS-purified B cell precursors in the presence of cancer conditioned media and observe production of macrophages which they claim indicates B cell transdifferentiation. However, the authors do not rule out the possibility of macrophage or macrophage precursors contaminating their FACS preparation. This is especially concerning because previous studies on B to macrophage transdifferentiation showed high (>80%) number of MΦ/B cell doublets when purifying B cell macrophages (Proc Natl Acad Sci U S A. 2017 May 16; 114(20): E3954–E3963). Additionally, the expression levels of CD11b and F4/80+ in the B cell macrophages purified in vivo

(Supplemental Figure 1B) are orders of magnitude lower than the ones observed for B cell macrophages grown in vitro (Figure 1C). These further suggest to this reviewer that the cultured B cell population might be due to contamination. This is a critical concern as most of the experiments in the manuscript are performed in macrophages differentiated in vitro. An easy way to address this concern the authors is through live imaging experiments tracking single cell differentiation of B cell precursors into macrophages in vitro. Alternatively, the authors could perform DNA barcoding experiments to demonstrate that macrophages and B cells in the culture share the cell of origin.

c) The strongest evidence for direct transdifferentiation of B cells into macrophages is the fact that B cell precursors from Mb1-cre:Csf1rflox/flox mice showed strong reductions in macrophage production in vitro (Fig. 4F). The manuscript will be considerably strengthened if the authors could show that the same phenotype takes place in vivo.

Lack of rigor in defining the different macrophage populations in FACS analyses

a) In most panels the authors define macrophages as CD11b+F4/80+ cells. This is not correct, F4/80 is widely expressed across myeloid populations of the bone marrow (main source of cells for most experiments) including monocytes, macrophages, and eosinophils whereas CD11b labels most myeloid cells (J Exp Med. 2011 Feb 14;208(2):261-71). The authors should rigorously exclude contamination of other cells in their samples prior claiming that the CD11b+F4/80+ cells detected are macrophages.

b) The F4/80+CD11b+CD19+ observed in vivo (Supplemental Figure S1B) express extremely low levels of both CD11b and F4/80. It is unlikely that these cells are bona fide macrophages and are clearly a completely different population than the CD11bbrightF4/80bright cells that the authors culture and analyze in most experiments. Much more detailed analyses are needed to demonstrate the identity of the CD11bdimF4/80dim cells. Importantly, in Supplemental Figure 1 the authors describe the B cell derived macrophages as CD19+ but the imagestream data shows that these cells are negative for CD19 (Figure 1H)

c) In line with the previous two comments: the authors quantify tumor associated macrophages (TAM, e.g. Fig. 1A-B, Fig. 1I) in mice bearing different tumors. It is not clear to this reviewer from which tissue the macrophages were purified. The methods state that the tumors were implanted subcutaneously, are the TAM shown in Fig. 1A-B skin macrophages? Do they express markers of skin macrophages (Nat Immunol. 2013 Oct; 14(10): 986-995)? What are the expression levels of CD11b and F4/80+ for the macrophages in the different panels? Do they correspond to the CD11bdimF4/80dim shown in Supplemental Figure S1B or to the CD11bbrightF4/80bright cells observed in cultures (e.g. Fig. 1C)? FACS plots for all these populations should be shown.

d) The authors profile the percentage of positive cells for numerous cell surface markers in CD79a+ (which they claim are B cell derived even though this is not shown) and CD79- macrophages in mice implanted with different tumors (Fig. S1E-G). Puzzlingly, they do not show the corresponding FACS plots, and the panels shown represent fractions of fractions of cells analyzed making the data essentially uninterpretable.

Novelty: previous studies already described the physiological transdifferentiation of B cell precursors into macrophages in response to inflammation and implicated M-CSFR in the B cell precursors as drivers of this process (Proc Natl Acad Sci U S A. 2017 May 16; 114(20): E3954-E396; J Immunol October 1, 1999, 163 (7) 3605-3611). It is thus expected that M-CSF producing cancer cells can induce a similar phenotype. It will greatly increase the novelty of the manuscript if the authors could demonstrate that the B cell derived macrophage have cancer related functions in vivo and in situ. This should be extremely simple by using the Mb1-cre:Csf1rflox/flox mice which should be unable to produce these macrophages in vivo.

Finally, the manuscript is very difficult to read due to the extraordinary amount of abbreviations used and the lack of details in the Figures and Figure legends.

Reviewer #3 (Remarks to the Author):

This is a well-executed body of research reporting on how tumours differentiate pre-B cells in to immune suppressive, tumour promoting macrophages.

Comments

Figure 1, panel C. Its unclear what animals were used here. Panel D, these cells don't look like macrophages based on nucleus shape and well as nuclear size-to-cytoplasm ratio. Do the authors have staining for F4/80? Would be helpful to include images of monocyte-derived conventional macrophage for comparison.

Figure 1, panel H. Please label each row within "WT" and "Mb1-cre-EYFP"

Figure 3A, Ki67 and BrDU uptake of about 40% in B-MF suggests quite a substantial rate of proliferation of these cells. Is this really likely to be the case in situ (in tumours)?

Figure 3C, the case for preferential uptake of Filipin by B-MF should be toned down as both cells types uptake Filipin quite well which is at odd with the black/white story painted in Figure 2D and E.

Reminiscent of data argues in Figure 3C, data presented in Figure 3E don't show much of a functional difference, again please tone down these differences in the text. Certainly, these data are statistically different from one another, but with such a high % efferocytosis, this will make little difference in situ as both cells type are very good at eating apoptotic cells.

Figure 3F appears to be not properly explained in the results section. Plus, the classic proliferation "shift" is not apparent in these data, perhaps I mis-understood their meaning?

Figure 3J. These are key experiments. Can the authors provide evidence of the location of injected B-MF within tumours? Also, was B-MF used as a comparator as well as B cells?

Indeed, this seems key data to be shown for their hypothesis – histology/confocal geographic location of labelled B-MF in the tumours of their model.

The term "anecdotal" (definition – "based on or consisting of reports or observations of usually unscientific observers") should be removed when referring to papers published in peer-reviewed journals of international repute.

Authors' point-by-point response to reviewer's comments

We thank the reviewers for their time and helpful suggestions. Below is our response to the comments raised, which whenever applicable, were addressed by performing additional experiments. Our manuscript has been substantially revised following the suggestions. Besides improving its clarity and adding missing controls, it contains new results to emphasize the key claim that cancers induce B-cell-to-macrophage differentiation.

Response to Reviewer 1: We thank the reviewer for finding our manuscript interesting and raising helpful questions, particularly on the relevance of B-MF to suppression of tumor rejection. The revised manuscript contains results of new experiments with two different but widely used tumor models (orthotopic B16-F10 melanoma and metastatic 4T1.2 breast cancer). The manuscript was also substantially revised to improve its clarity and, of course, to include your suggestions.

Reviewer 1, comment #1: *"One of my major concerns relates to how unique this population is for the tumor environment. The B-MF cells appear rare among the TIB, are these cells found in spleen or BM of non-tumor inoculated mice. Are they present outside of the tumor environment in the tumor inoculated mice? Analysis of the formation of these cells using a lineage tracer are done in panel II but then using a new tumor model as compared to 1A and SIC. In II mice without tumor obtain over 2% F480CD11b cells, higher levels than what is reported among TIB in Figure 1A".*

- Our answer: You are right that these cells appear to be quite rare. Our data do support that they are preferentially located in the tumor or tumor microenvironment, though we also did detect a small number of B-MF-like cells in naïve mouse spleen, LN and peritoneum, consistent with reports from others suggesting that they can be generated in unperturbed mice. To resolve this question, we performed a new experiment with ID8 ovarian cancer growing in peritoneum or s.c. MC38 tumor of B-cell tracer Mb1-EYFP mice showed (Fig.1E,F and Fig.S2A,B), B-MF were markedly increased in the tumor microenvironment. Although the B-MF are also increased in the primary tumors of almost every cancer type (except B16-F10 melanoma), the FACS values will differ depending on a cancer type as well because of the FACS processing of samples was done at different times. Thus, the conclusion on increase or decrease of cells are only made for the experiments performed at the same time. Of course, the results were reproduced at least in three independent experiments.

Reviewer 1, comment #1 (continued): *"There are experiments to this end presented in Figure S2. Here, however, it is unclear what "naïve" stand for. Wt mice? As the text is written it reads as if naïve would be mice not exposed to MC38. EYFP+ F4/80+CD11b^{inter/low} TAM were significantly increased in the tumor (Fig. S2)." If so, what organ was used for analysis of control mice, where is the statistical analysis? and why are not the CD19+ cells in the control mice YFP+? This needs to be resolved."*

- Our answer: We apologize for confusion. The "naïve" stands for "tumor-free" mice. The experiment was repeated using another tumor model, ID8 cancer in Mb1-EYFP mice. The revised manuscript contains the statistical analysis, and we state in the Results section states that: "... Compared to tumor-free (naïve) mice, peritoneum of tumor-bearing mice was significantly enriched in EYFP⁺ B-MF (Fig.1E,F and Fig.S2A,B) with upregulated expression of CD274 and TGFb/LAP (Fig.S2C), the two immunoregulatory factors In contrast, regardless of the tumor-bearing or naïve states of mice, these cells were only present at a small frequency in the spleen and LN (Fig.S2A,B). ..."

Reviewer 1, comment #1 (continued): “I would strongly recommend that the presence of B-MF cells in normal as well as tumor inoculated mice is carefully investigated to determine if this indeed represent a tumor specific population and if so, is it restricted to the tumor environment?”

- Our answer: As we noted above, we now state that B-MF are only generated in the tumor and tumor microenvironment (compare B-MF in tumor-bearing and naïve peritoneum as well in the spleen/LN of tumor-bearing and naïve mice, Fig.1E,F and Fig.S2A,B).

Reviewer 1, comment #2: “A second concern relates to the actual cell of origin. The data suggest that the cells are generated from progenitor cells. However, as the absolute majority are IgM+ in the absence of surrogate light chain expression (Figure S4). This strongly suggest that the cell of origin is an Immature B rather than IgM negative progenitors. Are BM progenitors from μ MT mice converted in the in vitro culture systems. It is also unclear to me how frequent CD19+ cells are in the tumor environment in the μ MT mice. If there are a substantial number of CD19+ cells in these mice they should all be progenitors and if not converted, would this not be conclusive evidence for that this is a unique ability of immature B-cells?”

- Our answer: Our in vitro experiments with highly FACS-purified B-cell subsets from the bone marrow (BM) indicate that cancers use pro-B, pre-B and immature B cells to generate B-MF (Fig.2E and Fig.S3F), particularly their Csf1r⁺CD93⁺ subsets. Please note that numbers of Csf1r⁺CD93⁺ subsets in pre-B cells and immature B cells (as well absolute numbers of pre-B cells and immature B cells) were markedly higher than that of pro-B cells (Fig.S7A and Fig.S7G), presumably explaining why we hardly detected B-MF in tumor-bearing uMT or JHT mice, where B-cell differentiation is blocked at the pro-B cell stage, (Fig.1B and C). It is also in concordance with a very low frequency of CD19⁺ B cells in the tumor tissue (~0.1% of total CD45⁺ immune cells) of tumor-bearing uMT mice as compared to that of tumor-bearing BALB/c mice (1-5% of total CD45⁺ immune cells) (Fig.S1D). Our results are consistent with our recent report (Ragonnaud et al., Cancer Research, 2019) - most BM B-cell precursors in the tumor and spleen of tumor-bearing mice are CD25⁺ pre-B cells as well immature B cells. New Fig.S1G,H shows that a large numbers of B cells clustered (presumably in TLS) in the primary tumor of mice with 4T1.2 and MC38 cancers.

Reviewer 1, comment #3: “The authors claim “4T1.2-CM, but not control medium, significantly upregulated F4/80 and CD11b in B cells from BM, but not from spleen, LN or PeC (Fig. 1C).” However, even though such data are shown for 70Z3 cells in panel 1F and G after 30 days of cultivation (The 30 day cultivation time should be indicated in the text), I cannot understand that Figure 1C contains data on either control medium or any statistical analysis of data from primary cells. This needs to be clarified and corrected”.

- Our answer: We apologize for not showing results of control media incubation. We used several different controls, such as cRPMI (a regular cell culture medium) as well B-cell-specific medium (they are described in the Methods section), they did not induce the generation of B-MF (see Fig.2D,G). Similarly, CM from B16-F10 melanoma cells failed to convert B-cell precursors or 70Z3 cells into B-MF (Fig.2G and Fig.S3I). We added the results of cRPMI, and the statistical analyses of Fig.2E, G are indicated in Fig.2F and Fig.S3H.
- You are correct that cancer converts B-MF only from BM B cells, but not from splenic, LN or PeC B cells (Fig.2E), if the tissues were from naïve (tumor-free) mice. In naïve mice, BM B-cell

precursors do not efficiently emigrate nor survive in those tissues (see our paper, Ragonnaud et al., Cancer Research, 2019; as well from others, Nie et al., J. Exp. Med., 2004). In mice with cancer, CD93⁺ B-cell precursors mobilize into the spleen and tumor (see the Section “Cancer mobilizes BMBP in the spleen to convert them to B-MF” and our recent paper, Ragonnaud et al., Cancer Research, 2019). As a result, cancer-CM readily generated B-MF from splenic B cells of tumor-bearing mice (Fig.S7C).

- As asked, the revised manuscript now indicates that: “BMBP and immortalized 70z/3 pre-B-cell line after 7 and 30 days of culture, respectively (Fig.2G and Fig.S3H-K)”

Reviewer 1, comment # 4: *"Figure 1D as well as S3B lack pictures of cells cultivated in "control media".*

- Our answer: The missing pictures for control cultures are now included in Fig.2D and in Fig.2G.

Reviewer 1, comment # 5: *"One of the most interesting findings is that immune suppressive function of the B-MF cells. This is studied in one experiment, using one tumor model and without any detailed analysis of the tumor environment. It would be important to know if the 300 000 B-MF transplanted home to the tumor. The ability of these cells to proliferate could potentially cause an additional expansion of this population in μ MT mice. How much of these cells are found in the tumor at the time of analysis? Are these levels comparable to what is observed in the tumor microenvironment in Wt mice? As the B-MF cells are CD45⁺, could that result in an apparent decrease in the T-cell fraction? Absolute number would be more relevant. As this is the most important part of this paper, I would recommend that this analysis is extended to involve more tumors as well as a much more detailed analysis of the tumor environment."*

- Our answer: We thank the reviewer for helpful suggestions. To expand our results on potential tumor-supporting role of B-MF, we performed additional in vivo experiments. Now as shown in Figures 4G-L, B-MF significantly increased tumor growth of B16-F10 melanoma in C57BL/6 mice and lung metastasis of 4T1.2 cancer implanted in the mammary gland of μ MT BALB/c mice. We linked this tumor-enhancing property to inhibition of tumor-infiltrating IFN γ ⁺ CD4⁺ T cells, which are known to have cytolytic antitumor activity (Xie et al., J. Exp. Med, 2010). B-MF did not affect cytolytic CD8⁺ T cells nor FoxP3⁺ Tregs in these mice. These results are consistent with our in vitro T cell suppression assay, where B-MF primarily inhibited IFN γ ⁺ CD4⁺ T cells. The molecular mechanism of this suppression is a topic of a different study.

Reviewer 1, Minor comments:

"1: I do not agree with the title of the paper as the cells generated do not appear to be conventional MF as they obtain unique transcriptional profile."

- Our answer: We modified the title as “Cancer coopts differentiation of B-cell precursors into macrophage-like cells”

"2: While it is clear from the data that the BM B-cells responds by upregulation of MF genes and a down regulation of B-lineage genes, it is rather unexpected that they to a large extend retain expression of CD79a (a gene by the way missing in the heat map in figure S4C) and IgM. As CD79 is a PAX5 target this would need to be discussed."

- Our answer: It should be noted that we “...capture these cells in “transition...”(see the section “Cancer induces B-cell transdifferentiation”). We also show results that “B cells gradually became

CD11b⁺F4/80⁺ while downregulating CD19 and some CD79 by 7-8 days of incubation in 4T1-CM, but not control cRPMI (Fig.2A,B, Fig.S3B and not depicted). After 14-days culture, the cells remained IgM⁺CD11b^{High}F4/80^{High} but further decreased CD19 and CD79a (Fig.S3C).”

“3: *What are the numbers in the FACS plots representing, average? representative exp? If average why not std inserted.*”

- Our answer: We apologize for missing legends’ information. They are now included. Almost every representative FACS plot has been quantified using independent samples with n=3-12 and reproduced at least three times. The Mean and SEM is included in every quantification. The numbers in the FACS plots are frequency (%) of the gated cells shown.

“4: *I would be useful to stress that this is a lineage tracer hence it does not report CD79 alpha expression but rather a history of CD79a expression where this model first is referred to.*”

- Our answer: Thank you. A modified sentence now states: “We also used Mb1-EYFP mice with or without peritoneal ID8 ovarian cancer to reveal history of CD79 expression in B-MF, as they express enhanced yellow fluorescent protein, EYFP, under promoter of B-cell exclusive Igα receptor (21).” (page 5)

“5: *Gene list in 2F appears misaligned.*”

- Our answer: Thank you. It is corrected.

=====
Response to Reviewer #2: We thank the reviewer for reviewing our manuscript and properly acknowledging the uniqueness of our study.

We respectfully disagree with the reviewer's statement that ".... *Unfortunately, the data presented does not rigorously support the central premise of the manuscript*". Although the B-cell-to-macrophage transdifferentiation has been reported to occur in mice albeit at very low levels, our manuscript for the first time reveals a biological relevance of this phenomenon. By experimenting with various murine tumor models in different mouse strains, we show that cancer markedly increase macrophage-like cells by way of transdifferentiation from B cells to promote cancer progression and metastasis. We also show that unlike recently reported biphenotypic pro-B cells (CD19⁺B220⁺CD16/32⁺⁺CD11b⁺) with non-rearranged B-cell receptor (BCR) genes, cancer-induce B-MF are derived from CD93⁺Csf1r⁺ subsets of the bone marrow (BM) pro-B cell, pre-B cells and immature B cells.

We agree with the reviewer that "*Demonstration of transdifferentiation: the central claim of the manuscript is that cancer reprograms a rare subset of B cell precursors into macrophages. This claim must be rigorously supported. Studies claiming transdifferentiation of hematopoietic cells were later shown to be due to cell fusion when more rigorous and well-controlled experiments were performed (e.g. Science. 2002;297:1299, Nature. 2003 Apr 24;422(6934):897-901, Nat Med. 2004 May;10(5):494-501). Further, cell fusion is enhanced by inflammation (Nat Cell Biol. 2008 May;10(5):584-92, Nat Cell Biol. 2008 May;10(5):575-83). The authors data supporting that B cell precursors give rise to macrophages is based on the identification of CD11b⁺F4/80⁺ cells within the CD19⁺ gate, that cultured B cell precursors generate macrophages in vitro, that B cell macrophages are lineage traced after transfer of B*

cell precursors from the Mb1-cre:EYFP mice, and that Mb1-cre:Csf1rflox/flox mice have impaired capacity to produce B macrophages. There are significant limitations and missing controls with these experiments.”

- Our answer: To support our central claim that the B-MF are derived from transdifferentiation and are not mistaken interpretations of trogocytosis or cell fusion, the revised manuscript contains results of new in vitro and in vivo experiments (see “Cancer induces B-cell transdifferentiation”). We exclude trogocytosis or cells fusion using a traceable and widely-utilized system of mice expressing pan-hematopoietic cell marker isotypes, CD45.1 and CD45.2. First, mixture of highly-FACS purified CD45.2⁺ Mb1-EYFP⁺ B-cell precursors and BM CD45.1⁺ monocytes were differentiated into B-MF and macrophages, respectively, in 4T1.2-CM. While only a small fraction (1-3%) of cells co-expressed CD45.2 and CD45.1 (presumably a result of trogocytosis or cell fusion), the majority of B-MF and Mo-MF only expressed their respective single alloantigen, CD45.2 or CD45.1 (Fig.2D), implying that most in vitro-generated B-MF are not derived from trogocytosis/cell fusion. We also implanted CD45.2⁺ EYFP⁺ B-cell precursors into congenic tumor-bearing mice expressing CD45.1. Again, a very small number of host and donor cells co-expressed CD45.1 and CD45.2/EYFP (Fig.2H,I and Fig.S4E), presumably again due to trogocytosis/cell fusion. However, the majority of EYFP⁺ B-MF did not express the host mouse CD45.1 (Fig.2I). Together with the results that the cells were highly pure (FACS-purified, Lin⁻ (CD11b, F4/80, GR1, Ly6G, Ly6C, TER119, CD49b, CD4, CD8, CD11c) >99% pure B cell precursors) and that only a small cell subsets (such as CD93⁺Csf1r⁺) within BM pro-B, pre-B and immature B cells, but not mature B cells, can generate B-MF, we conclude that B-MF are not derived from cell fusion or trogocytosis. This conclusion is also supported by the unique transcriptional, metabolic, and functional features of B-MF, which differ from that of BM monocyte-derived macrophages.

Reviewer 2, comment a: "a) The authors should demonstrate that the B cell macrophages are generated by transdifferentiation and not via cell fusion. The easiest way to examine this will be to transfer Mb1-YFP⁺ precursors into Ubc-gfp (or other pan fluorescent protein reporter) mice and demonstrate that the B cell macrophages are GFP⁻."

- Our answer: As discussed above, we excluded the primary role of cell fusion or trogocytosis in the B-MF generation using in vitro and in vivo experiments with CD45.2⁺ Mb1-YEFP⁺ B-cell precursors either mixed with CD45.1⁺ monocytes/macrophages or transferred into tumor-bearing mice expressing CD45.1.

Reviewer 2, comment b: "b) The authors culture FACS-purified B cell precursors in the presence of cancer conditioned media and observe production of macrophages which they claim indicates B cell transdifferentiation. However, the authors do not rule out the possibility of macrophage or macrophage precursors contaminating their FACS preparation. This is especially concerning because previous studies on B to macrophage transdifferentiation showed high (>80%) number of MΦ/B cell doublets when purifying B cell macrophages (Proc Natl Acad Sci U S A. 2017 May 16; 114(20): E3954–E3963). Additionally, the expression levels of CD11b and F4/80⁺ in the B cell macrophages purified in vivo (Supplemental Figure 1B) are orders of magnitude lower than the ones observed for B cell macrophages grown in vitro (Figure 1C). These further suggest to this reviewer that the cultured B cell population might be due to contamination. This is a critical concern as most of the experiments in the manuscript are performed in macrophages differentiated in vitro. An easy way to address this concern the authors is through live imaging experiments tracking single cell differentiation of B cell precursors into

macrophages in vitro. Alternatively, the authors could perform DNA barcoding experiments to demonstrate that macrophages and B cells in the culture share the cell of origin."

- Our answer: We acknowledge the concern that there may be contamination, but present more clear data that we find robust and convincing, to eliminate this phenotype as an artifact of cell sorting (either doublets or myeloid cells). First, our cells were Lin⁻ (including markers that would be present on macrophage precursors) and >99% pure B-cell precursors (see cell purity results of our highly FACS-purified B-cell precursors in Fig.S3A). We did everything within currently available methodology to exclude contaminating myeloid precursor cells. In addition, using B-cell lineage tracer mice (Mb1-EYFP) as well B-cell precursors from mice with B-cell-specific Csf1R deficiency, we show that loss of B-cell progenitors or B-cell specific impairment Csf1r block the generation of B-MF. Lastly, B-MF exhibit unique transcription, metabolic, and functional features that are different from macrophages generated the same way from the bone marrow monocytes. Lastly, B-MF expressed IgM, IgD and CD20 (Fig.1B, Fig.S1 and Fig.S3C).

Reviewer 2, comment c: “*c) The strongest evidence for direct transdifferentiation of B cells into macrophages is the fact that B cell precursors from Mb1-cre:Csf1rflox/flox mice showed strong reductions in macrophage production in vitro (Fig. 4F). The manuscript will be considerably strengthened if the authors could show that the same phenotype takes place in vivo.*”

- Our answer: Thank you, we agree with the reviewer’s statement that the experiment with Mb1-cre:Csf1r^{flox/flox} mouse cells is our strongest evidence. For unknown reasons, we are having trouble with breeding Mb1-cre:Csf1r^{flox/flox} mice and having only a few mice of different ages due to these breeding issues unfortunately precludes an in vivo tumor study for a timely response. However, we present additional experimental evidence in the manuscript to support our claim, including in vivo conversion of B-MF from Mb1-EYFP B-cell lineage tracer mice, and hope that though it not the exact experiment suggested, the reviewer will find these experiments sufficiently convincing.

Reviewer 2 commented: “*Lack of rigor in defining the different macrophage populations in FACS analyses*

a) In most panels the authors define macrophages as CD11b+F4/80+ cells. This is not correct, F4/80 is widely expressed across myeloid populations of the bone marrow (main source of cells for most experiments) including monocytes, macrophages, and eosinophils whereas CD11b labels most myeloid cells (J Exp Med. 2011 Feb 14;208(2):261-71). The authors should rigorously exclude contamination of other cells in their samples prior claiming that the CD11b+F4/80+ cells detected are macrophages.”

- Our answer: We respectfully disagree with this comment of reviewer 2. Although we are not debating or excluding that our B-MF could also become other myeloid cells, as did pro-B cells after artificial/forced manipulation of genes, FACS analysis of B-MF shown in Fig.S5 indicates that they are not PMN/granulocytes or DCs. Although we term B-MF as macrophages based on commonly accepted markers (CD11b^{High}F4/80^{High}), microscopy analyses and transcription profiling as well scRNAseq suggest that B-MF are macrophages. Our B-MF also phagocytized apoptotic cancer cells and adhered to plastic – features linked to macrophages or macrophage-like cells. That said, we did also modify the title of the manuscript to “Cancer coopts differentiation of B-cell precursors into **macrophage-like** cells

Reviewer 2 commented: “b) The F4/80+CD11b+CD19+ observed in vivo (Supplemental Figure S1B) express extremely low levels of both CD11b and F4/80. It is unlikely that these cells are bona fide macrophages and are clearly a completely different population than the CD11bbrightF4/80bright cells that the authors culture and analyze in most experiments. Much more detailed analyses are needed to demonstrate the identity of the CD11bdimF4/80dim cells. Importantly, in Supplemental Figure 1 the authors describe the B cell derived macrophages as CD19+ but the imagestream data shows that these cells are negative for CD19 (Figure 1H)”

- Our answer: We apologize for confusing figures. New Fig.S1A as well Fig.S2A clearly show that our cells are CD11b^{High} and F4/80^{High}. The CD19 negativity in the imagestream data is consistent with our in vitro B-MF generation results, which show that CD19 is gradually lost while cells become CD11b^{High} and F4/80^{High} (see Fig.2A and Fig.S3B,C).

Reviewer 2 commented: “c) In line with the previous two comments: the authors quantify tumor associated macrophages (TAM, e.g. Fig. 1A-B, Fig. 1I) in mice bearing different tumors. It is not clear to this reviewer from which tissue the macrophages were purified. The methods state that the tumors were implanted subcutaneously, are the TAM shown in Fig. 1A-B skin macrophages? Do they express markers of skin macrophages (Nat Immunol. 2013 Oct; 14(10): 986–995)? What are the expression levels of CD11b and F4/80+ for the macrophages in the different panels? Do they correspond to the CD11bdimF4/80dim shown in Supplemental Figure S1B or to the CD11bbrightF4/80bright cells observed in cultures (e.g. Fig. 1C)? FACS plots for all these populations should be shown.”

- Our answer: We apologize for confusion. The missing information is included in the revised manuscript: The TAM are tumor-associated macrophages isolated from the primary tumors of mice, such as s.c. growing MC38, 4T1.2 and EMT6 cancers and in the tumor microenvironment (peritoneum) of ID8 and Mogp cancers growing in the peritoneum.
- The characterization of the skin macrophages or other resident macrophages we also find intriguing, but feel it to be topic of a different study, as the focus of this manuscript is not to fully characterize these different types of macrophages, but to present a novel mechanism that cancer co-opts to convert a lymphoid cell to have myeloid /macrophage properties. Per manuscript publication guidance and limits, we now provide as many as possible representative FACS plots and gating strategies to clarify how populations we evaluated were assayed.

Reviewer 2 commented: “d) The authors profile the percentage of positive cells for numerous cell surface markers in CD79a+ (which they claim are B cell derived even though this is not shown) and CD79- macrophages in mice implanted with different tumors (Fig. S1E-G). Puzzlingly, they do not show the corresponding FACS plots, and the panels shown represent fractions of fractions of cells analyzed making the data essentially uninterpretable.”

- Our answer: We did not mean to appear to withhold FACS plots, we had only hoped to not overwhelm the readers with too many. We apologize if that made some of the interpretations challenging, and provide the FACS plot and gating strategy in Fig.S1A. We feel this now clearly demonstrates expression of F4/80, CD11b, Filipin, CD206, CD20 and CD19 in CD79+ and CD79- cells. Also, Fig.S1C shows FACS data histogram on expression levels of CD19, CD20 and IgM in B-MF. We hope these provide more clarity to the panels previously presented.

Reviewer 2 commented: “Novelty: previous studies already described the physiological transdifferentiation of B cell precursors into macrophages in response to inflammation and implicated M-CSFR in the B cell precursors as drivers of this process (*Proc Natl Acad Sci U S A.* 2017May 16; 114(20): E3954–E396; *J Immunol* October 1, 1999, 163 (7) 3605-3611). It is thus expected that M-CSF producing cancer cells can induce a similar phenotype. It will greatly increase the novelty of the manuscript if the authors could demonstrate that the B cell derived macrophage have cancer related functions in vivo and in situ. This should be extremely simple by using the *Mb1-cre:Csf1rflox/flox* mice which should be unable to produce these macrophages in vivo.

Finally, the manuscript is very difficult to read due to the extraordinary amount of abbreviations used and the lack of details in the Figures and Figure legends.”

- Our answer: Although the B-cell-to-macrophage transdifferentiation phenomenon has been proposed by others for some time, the biological relevance of the phenomenon remained unclear. The recent study mentioned by reviewer (which we now discuss in more details our revised manuscript) further underscores importance of the findings from our two groups. However, the cells that differentiate into macrophages in that paper (*Proc Natl Acad Sci U S A.* 2017 May 16; 114(20): E3954–E396) differ significantly from our B-cell precursors. First, their cells are biphenotypic pro-B cells (CD19⁺B220⁺CD16/32⁺⁺CD11b⁺) without rearranged B-cell receptor (BCR) genes and do not express CD79b. These precursors appear, but do not mobilize into spleen. In contrast, our cells are Lin⁻ (CD93⁺Csf1R⁺CD79a⁺CD19⁺) B-cell subsets of BM pro-B, pre-B and even immature B cells (indicating a rearranged BCR) and they infiltrate into the spleen and tumor. Importantly, the loss of pre-B and immature B cells in uMT and JHT mice impairs the generation of B-MF. For the first time we show a clinical relevance of the transdifferentiation of B cells. Using two different, widely-accepted cancer models, we link B-MF to promotion of tumor growth and lung metastasis. To do this, B-MF appear to primarily target IFN γ ⁺CD4⁺ T cells (but not CD8⁺ T cells or FoxP3⁺ Tregs). We herein provide many novel findings, and for the first time, show biological relevance of the phenomenon, i.e. link the B-cell-to-macrophage transdifferentiation to cancer. Although M-CSF can be expected to be involved in the generation of B-MF, we come to our conclusion via screening cancer CM.
- We apologize that the manuscript was difficult to read. We hope that our revised manuscript will be easier to follow, as we tried to frequently explain abbreviations, but are unfortunately used to compress word numbers to satisfy word limits.

=====
Response to Reviewer 3: We thank the reviewer finding our manuscript “... is a well-executed body of research reporting on how tumours differentiate pre-B cells in to immune suppressive, tumour promoting macrophages”

Reviewer #3 commented: “ Figure 1, panel C. Its unclear what animals were used here. Panel D, these cells don’t look like macrophages based on nucleus shape and well as nuclear size-to-cytoplasm ratio. Do the authors have staining for F4/80? Would be helpful to include images of monocyte-derived conventional macrophage for comparison. Figure 1, panel H. Please label each row within “WT” and “Mb1-cre-EYFP”

- Our answer: We added additional panels (Fig.1D) and new Figure (Fig.S5B) to also show images of BM monocyte-derived macrophages and primary peritoneal macrophages. B-MF look very similar to these two types of macrophages and different from B cells. The peritoneal macrophages were taken from naïve mice as noted in Fig.S5B legend.

Reviewer #3 commented: " *Figure 3A, Ki67 and BrdU uptake of about 40% in B-MF suggests quite a substantial rate of proliferation of these cells. Is this really likely to be the case in situ (in tumours)? Figure 3C, the case for preferential uptake of Filipin by B-MF should be toned down as both cells types uptake Filipin quite well which is at odd with the black/white story painted in Figure 2D and E. Reminiscent of data argues in Figure 3C, data presented in Figure 3E don't show much of a functional difference, again please tone down these differences in the text. Certainly, these data are statistically different from one another, but with such a high % efferocytosis, this will make little difference in situ as both cells type are very good at eating apoptotic cells.*"

- Our answer: You are right that "... Ki67 and BrdU uptake of about 40% in B-MF ..." (now Fig.4A. and Fig.S6A,B). Please note that this is at the day 7 of B-MF generation, and the cycling decreases when maintained for longer in culture (Fig.S3C). The data serve to show differences between B-MF and Mo-MF differentiated in the same condition. We do not have a definitive data on Ki-67 or BrdU uptake (proliferative state of macrophages) in vivo, a topic of a different study. We have toned down our statements in the Results section as you suggested to state "Although the two macrophages phagocytized fluorochrome-labeled apoptotic cancer cells (Fig.4B, C) and contained elevated levels of cellular cholesterol (Fig.4D, E), both these features were upregulated in B-MF compared to Mo-MF per cell-to-cell comparisons (Fig.4C, E)".

Reviewer #3 commented: " *Figure 3F appears to be not properly explained in the results section. Plus, the classic proliferation "shift" is not apparent in these data, perhaps I mis-understood their meaning?*"

- Our answer: We apologize for not properly explaining Fig.3F. Therefore, in revised manuscript we expanded the explanation of the results of the T cell suppression assay. As we often observed in many our T cell suppression assays (see our papers, Ragonnaud et al., Cancer Research, 2019; Bodogai et al., Cancer Research, 2015; Bodogai et al., Cancer Research, 2013; Olkhanud et al., Cancer Research, 2011), the dilution of CFSE label (i.e. representing cell proliferation) often results in profiles like this (see profiles of non-activated T cells vs T cells stimulated with anti-CD3/CD28 Abs in Fig.S6C,D). In the revised manuscript, complete quantification of proliferated cells are now shown in Fig.4F and their representative FACS histograms in Fig.S6C,D).

Reviewer #3 commented: " *Figure 3J. These are key experiments. Can the authors provide evidence of the location of injected B-MF within tumours? Also, was B-MF used as a comparator as well as B cells? Indeed, this seems key data to be shown for their hypothesis – histology/confocal geographic location of labelled B-MF in the tumours of their model.*"

- Our answer: To clarify these questions we have included a new figure (Fig.S2B) showing quantification of EYFP⁺ B-MF present in the primary tumor of mice with MC38, which is consistent with similar results in Fig.2I and Fig.S4C (showing conversion of transferred EYFP⁺ B cells in the tumor microenvironment (peritoneum) of EYFP⁺ mice with ID8 cancer) and Fig.1E and Fig.S2B (B-MF cells in peritoneum of mice with peritoneal ID 8 tumor) and Fig.1B,C and

Fig.S1 (showing presence of B-MF in various primary tumors of mice). In new Fig.1D and Fig.S1G-H, we also show confocal microscopy results of immunohistochemistry stained primary tumors of mice with 4T1.2 and MC38 cancers, where we show presence of B-MF – like cells.

Reviewer #3 commented: " *The term "anecdotal" (definition – "based on or consisting of reports or observations of usually unscientific observers") should be removed when referring to papers published in peer-reviewed journals of international repute.*"

- Our answer: Thank you for your advice: we have removed the term "anecdotal".

REVIEWER COMMENTS

Reviewer #1 (Remarks to the Author):

Comment revised manuscript:

While the revised version of the paper is improved, some of my concerns are still not addressed and some new information raise new questions and concerns.

Specifically:

I do not believe that the authors has addressed concerns raised in my previous comment 5. The ability of the B-MF to prevent an optimal anti-tumor response, as in this version of the paper proposed, via selective inhibition of the IFN γ +CD4+ population, is absolutely central for the paper. Therefore I suggested that the distribution and presence of the transplanted in vitro generated B-MF should be investigated. This motivated by that it is unclear how these cells behave when transplanted to B-cell deficient mice. Are they expanding? Reverting into more conventional B-cells? Homing to the tumor? This is of large importance for the conclusion of this paper. And as the authors correctly state "because the lack of B cells retards progression of these tumors 6." a most relevant control would be transplantation of conventional B-cells in the same tumor settings. This to show a unique role for the B-MF cells in the tumor-host interaction.

In my original comment 5, I suggested that the data on T-cell populations should not be calculated as % CD45 cells as any infiltration of transplanted B-MF cells would cause an alteration in the CD45+populations possibly without changing the actual number of CD4 cells in the tumor. In the original version of this paper the authors claimed that they observed a change in the frequency of CD4+ cells (Figure 3H 1st version of the paper). This claim is now, as far as I understand, shown to be incorrect (Figure S6F) and replaced by the statement that there is a shift in the frequency of IFN γ CD4+ cells. However, the issue with analysis of frequency of CD45+ cells remains in panel S6F and G. Furthermore, would not the fact that there are no changes in the frequency of T-cell populations indicate that the transplanted B-MF cells are few in the tumor?

Therefore, I cannot see that the authors prove a unique role for the B-MF in the tumor-host interaction, a central aspect in this paper.

The response to my minor comment 4 is not easy to understand. As far as I can see and understand from the materials and methods, the authors use a Mb1-cre to activate a Rosa46 EYFP reporter. In their response and the new version of this manuscript the authors claim that " as they express enhanced yellow fluorescent protein, EYFP, under promoter of B-cell exclusive Ig α receptor (21). " and in the manuscript " We also used 121 Mb1-EYFP mice with or without peritoneal ID8 ovarian cancer to reveal history of CD79 122 expression in B-MF, as they express enhanced yellow fluorescent protein, EYFP, under the promoter of B-cell exclusive Ig α receptor 24." I find this most confusing.

Minor comments:

It is generally difficult to follow the gating strategies used. It would simplify if it was indicated in the figures.

In Figure S3E the authors use cells from Rag-GFP mice. Why are they still GFP+? Why did the authors use a Rag reporter and what Rag reporter was used, I cannot find this in information in the M&M section.

Reviewer #2 (Remarks to the Author):

The authors have addressed my key concerns.

Reviewer #3 (Remarks to the Author):

I AM HAPPY WITH AUTHORS REPLIES

Authors' point-by-point response to reviewer's comments

We thank the reviewers for finding our revised manuscript satisfactory. Below is our response to the comments raised by reviewer #1, which whenever applicable, were addressed by performing additional experiments.

Reviewer 1, comment #1: *“While the revised version of the paper is improved, some of my concerns are still not addressed and some new information raise new questions and concerns.*

Specifically:

I do not believe that the authors has addressed concerns raised in my previous comment 5. The ability of the B-MF to prevent an optimal anti-tumor response, as in this version of the paper proposed, via selective inhibition of the IFN γ +CD4 $^+$ population, is absolutely central for the paper. Therefore I suggested that the distribution and presence of the transplanted in vitro generated B-MF should be investigated. This motivated by that it is unclear how these cells behave when transplanted to B-cell deficient mice. Are they expanding? Reverting into more conventional B-cells? Homing to the tumor? This is of large importance for the conclusion of this paper. And as the authors correctly state “because the lack of B cells retards progression of these tumors 6.” a most relevant control would be transplantation of conventional B-cells in the same tumor settings. This to show a unique role for the B-MF cells in the tumor-host interaction.”

- Our answer: As asked, we performed additional experiments to address the concerns raised, including evaluation of eFluor450-labeled B-MF after transfer into tumor-bearing mice. The results, which are now shown in Suppl. Fig. 6I and J and described in the Results section, indicate that the overwhelming majority of the transferred B-MF retain CD11b $^+$ F4/80 $^+$ expression (>98%, Fig.S6J), thus there is no indication that these cells reverted back into B cells and thereby supported cancer progression and metastasis. This result is fully consistent with our in vitro results where B-cell-to-macrophage differentiation was unidirectional and B-MF did not revert back to B cells (see Fig.2A,B and Fig.S3B,C). This conclusion is also consistent with our new data (see Fig.S6H), which compares tumor-infiltrating T cells in mice with 4T1.2 cancer after transfer of equal numbers of B-MF or follicular B cells. Unlike B-MF, which primarily downregulated numbers and frequency IFN γ +CD4 $^+$ T cells (but not CD8 T cells and Tregs, Fig.4J-M and Fig.S6F and G), transferred B cells significantly increased IL10 $^+$ CD4 $^+$ T cells (numbers and frequency) and decreased frequency of cytolytic CD8 T cells (Fig.S6H), which is also consistent with our previous reports that B cells primarily target CD8 and Tregs to promote metastasis. Overall, these results suggest that the cancer-promoting function of B-MF does not involve their reversal to B cells.

Reviewer 1, comment #2: *“In my original comment 5, I suggested that the data on T-cell populations should not be calculated as % CD45 $^+$ cells as any infiltration of transplanted B-MF cells would cause an alteration in the CD45 $^+$ populations possibly without changing the actual number of CD4 cells in the tumor. In the original version of this paper the authors claimed that they observed a change in the frequency of CD4 $^+$ cells (Figure 3H 1st version of the paper). This claim is now, as far as I understand, shown to be incorrect (Figure S6F) and replaced by the statement that there is a shift in the frequency of IFN γ CD4 $^+$ cells. However, the issue with analysis of frequency of CD45 $^+$ cells remains in panel S6F and G. Furthermore, would not the fact that there are no changes in the frequency of T-cell populations indicate that the transplanted B-MF cells are few in the tumor? Therefore, I cannot see that the authors*

prove a unique role for the B-MF in the tumor-host interaction, a central aspect in this paper. ”

- Our answer: As asked, absolute numbers and frequency of T cells are now included in the manuscript (Fig.4J-M, Fig.S6F,G, and Fig.S6H) and the revised manuscript states that “the B-MF transfer significantly decreased frequency and numbers of IFN γ -expressing CD4⁺ T cells in both cancer models (Fig.4J-M)” (see the Results section). Although this was not a key aspect of the paper, together with our in vitro T cell suppression assay (Fig.4F and Fig.S6C-E) these results support our conclusion that B-MF promote cancer escape at least in part by downregulating IFN γ -expressing CD4⁺ T cells.

Reviewer 1, comment #3: *“ The response to my minor comment 4 is not easy to understand. As far as I can see and understand from the materials and methods, the authors use a Mb1-cre to activate a Rosa46 EYFP reporter. In their response and the new version of this manuscript the authors claim that ” as they express enhanced yellow fluorescent protein, EYFP, under promoter of B-cell exclusive Ig α receptor (21). ” and in the manuscript “ We also used 121 Mb1-EYFP mice with or without peritoneal ID8 ovarian cancer to reveal history of CD79 122 expression in B-MF, as they express enhanced yellow fluorescent protein, EYFP, under the promoter of B-cell exclusive Ig α receptor 24.” I find this most confusing.”*

- Our answer: We apologize for the typo and the confusion. A revised sentence states “We also evaluated B-MF in Mb1-Cre/Rosa-EYFP crossed (Mb1-EYFP) mice with or without peritoneal ID8 ovarian cancer, where Mb1-dependent Cre-recombinase causes B-cell-exclusive expression of EYFP (enhanced yellow fluorescent protein) ²⁴.”(see the Results Section).

Reviewer 1, comment #4: *“ Minor comments: It is generally difficult to follow the gating strategies used. It would simplify if it was indicated in the figures.*

In Figure S3E the authors use cells from Rag-GFP mice. Why are they still GFP+? Why did the authors use a Rag reporter and what Rag reporter was used, I cannot find this in information in the M&M section”.

- Our answer: Due to space limit in primary Figures, we only referenced to gating strategies in the text (the Results section). To make easier to follow, we now also referenced the relevant gating strategies in the legends of Figure and Suppl. Figure legends (see for example, legends for Figs. 1, 2, and 5, marked Red).
- The reference to RAG2-GFP mice (which express bacterial artificial chromosome modified GFP instead of RAG2, a gift of Dr. Michael Nussenzweig (Howard Hufhes Medical Institute, NY, NY) is included in MM section. We used these mice to FACS-purify BM pro and pre-B cells (based on GFP), which then were converted into B-MF after 7-day culture in 4T1.2-CM. Note that after 7-days of conversion, GFP signal was detectable in B-MF (presumably due to longer stability of GFP) despite shutdown of RAG expression (as no RAG mRNA was detected in RNAseq).

REVIEWERS' COMMENTS

Reviewer #1 (Remarks to the Author):

I wish to thank the authors for completing additional experiments and for adding the new data to the manuscript. I know think it is possible for an expert reader to validate the data presented.